Journal of Data-centric Machine Learning Research (2025)          Submitted 10/24; Revised 08/25; Published 09/25

# DecordFace: A Framework for Degraded and Corrupted Face Recognition

**Surbhi Mittal**                                                                          MITTAL.5@IITJ.AC.IN
*Department of Computer Science*
*IIT Jodhpur*

**Rishi Dey Chowdhury**                                                          MB2320@ISICAL.AC.IN
*Department of Statistics*
*ISI Kolkata*

**Mayank Vatsa**                                                                       MVATSA@IITJ.AC.IN
*Department of Computer Science*
*IIT Jodhpur*

**Richa Singh**                                                                           RICHA@IITJ.AC.IN
*Department of Computer Science*
*IIT Jodhpur*

**Reviewed on OpenReview:** *https://openreview.net/forum?id=XJoY2EWROn*

**Editor:** Sergio Escalera

## Abstract

Face recognition (FR) models have become an integral part of day-to-day activities involving surveillance and biometric verification. While these models perform remarkably well in constrained settings, their performance is limited in the presence of certain challenging covariates. One such covariate is the presence of unforeseen image degradations and corruptions. These degradations, which inevitably occur during image acquisition, transmission, or storage, substantially impact real-world applicability. In order to analyze the performance of FR systems in these scenarios, we provide the first-ever Degraded and Corrupted Face Recognition (DecordFace) framework to evaluate the robustness of FR models. Corrupted versions of multiple standard datasets are created, and experiments are performed using more than 3.6 million corrupted face images with over 25 recognition models with different architectures and backbones, using 16 corruptions at 5 severity levels. For quantitative estimation of the impact of corruption, we introduce two novel evaluation metrics, error-based mVCE and embedding-based mCEI. Using these metrics and a cohort of FR models, we conduct a detailed analysis of model robustness under different model and input parameters. We observe a severe drop in the performance of models for unconstrained face recognition with performance errors over 20% across different corruptions. The performance of model variants with shallow backbones is observed to suffer even more. The code for the DecordFace framework can be accessed at `https://github.com/IAB-IITJ/DecordFace`.

**Keywords:**   Face recognition, Image Quality, Corruption, Framework, Metrics

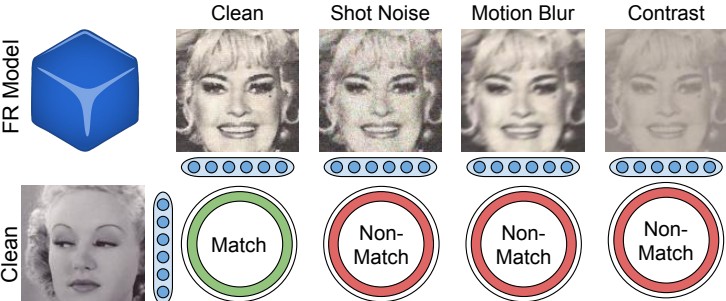

Figure 1: The performance of state-of-the-art FR models degrades under image corruptions. Predictions are shown for the Adaface iResNet100 model (Kim et al., 2022) trained on MS1MV2 (Deng et al., 2019) dataset. The image pair belongs to the AgeDB-*decord* (Moschoglou et al., 2017) set.

## 1 Introduction

Deep learning-based technologies have resulted in face recognition models becoming an indispensable part of our everyday interactions. They have also been widely used for security and surveillance-based applications from airports to mobile devices. Therefore, it has become necessary to ensure their robustness, especially in safety-critical applications. The applications of FR often involve the acquisition of face images in unconstrained environments. Despite the high accuracies of FR models in constrained settings (Schroff et al., 2015; Majumdar et al., 2017), the performance of FR models has been observed to degrade under unconstrained settings (Meng et al., 2021b; Kim et al., 2022).

Since variations in environmental conditions are inherent in the real world, they can lead to the incorporation of common corruptions and degradations in images, such as brightness or blur. Further, the acquired images are transmitted for storage or evaluation. The process of transmission and storage can further introduce noise and compression artifacts in images. Addressing these corruptions and degradations is crucial for ensuring accurate and reliable face recognition. In the literature, the impact of degradations and corruptions on face images has been studied, where researchers have observed a drop in the performance of FR models (Grm et al., 2017; Goswami et al., 2018) (Fig. 1).[1]

Further, studies analyzing the impact of corruptions have been conducted in other domains, such as object classification, where image corruptions have been shown to heavily degrade performance (Hendrycks and Dietterich, 2019). The impact of corruptions is thoroughly analyzed in these works, and the performance drop has been attributed to a distribution shift in the data (Hendrycks and Dietterich, 2019). However, no such study has been conducted in the domain of face recognition, and there is no existing framework that evaluates state-of-the-art FR models on datasets containing age, pose, and other important covariates. In this work, we present a thorough framework that focuses on 16 different corruptions under noise, blur, light and color, distortion, compression, and occlusion, each at five levels of severity (refer Fig. 2).

---

1. *We use the terms degradations and corruptions interchangeably in this work.*

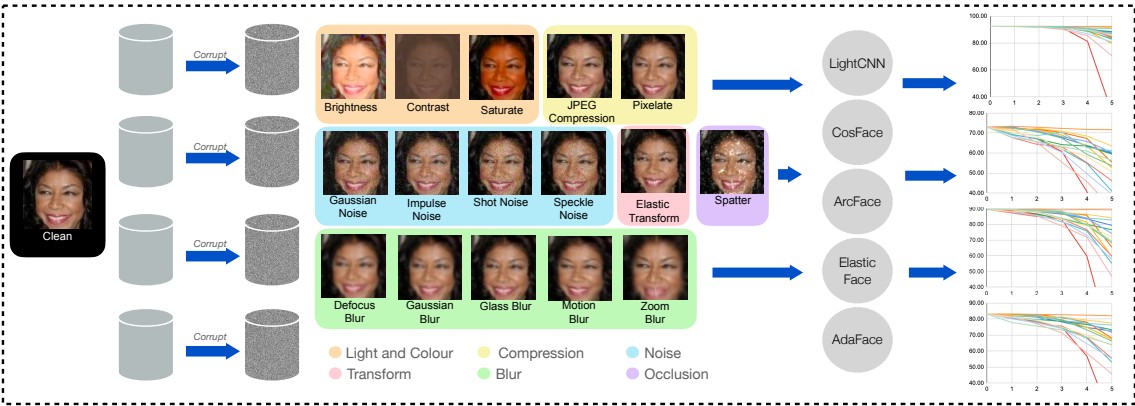

Figure 2: The DecordFace framework corrupts multiple FR datasets with 16 image corruptions at 5 severity levels. The image corruptions can broadly be classified into the categories of light and color, compression, noise, transform, blur, and occlusion. On evaluation using multiple FR models, the verification performance for different corruptions decreases with increasing corruption severity. The samples are shown at severity level 3, and the clean image belongs to the CPLFW dataset (Zheng and Deng, 2018).

In the proposed Degraded and Corrupted Face Recognition (DecordFace) framework, we create corrupted versions of 5 popular FR datasets used for face verification, including the AgeDB (age variation), CALFW (age variation), CFP-FP (pose variation), CPLFW (pose variation) and the large-scale IJB-C datasets. A total of 80 corruptions with 16 corruption types, each at five severity levels, are utilized to corrupt the face images. In the DecordFace framework, we analyze the performance of over 25 popular face recognition models. These models include the popular FR models with variations in the depth of model backbones. Two metrics are proposed to quantify the effect of corruptions on model performance- the mean Verification Corruption Error (mVCE) and mean Corruption Embedding Invariance (mCEI). While the mVCE metric evaluates the verification performance of a given model in the presence of corruption, mCEI focuses on evaluating the influence on the feature embedding space. Detailed analysis across factors of corruption type, the training dataset, model design, and influence on model fairness showcases the relevance of the proposed framework. The key observations via DecordFace include,

- The performance drop in the vast majority of models occurs for noise-based and blur-based corruptions, including state-of-the-art quality-aware methods such as Adaface (Kim et al., 2022). We further observe a significant dip in model performance due to a change in contrast, contrary to observations made in previous work (Grm et al., 2017).

- The performance of shallow model backbones, such as those with 18 and 34 layers, show a significant drop in model performance even at lower corruption severity levels.

This performance drop is especially alarming since these models perform well on non-corrupted face image pairs.

- On evaluating model performance using dataset-defined 'gender' subgroups, it is observed that the model performance drops significantly more for the female subgroup for all corruptions.

Through *DecordFace*, we aim to provide a robust framework for comparison of FR algorithms in the presence of distribution shifts caused by common corruptions. The framework would act as a good valuation step during training, leading to robust models. To support the relevance of our diagnostic framework, we also evaluate on two datasets with naturally occurring degradations, CelebA Blurry and a distance-based subset of D-LORD, where we observe that model performance trends align with those observed under synthetic corruptions.

## 2 Related Work

The impact of image quality and data distribution shifts, where the test data statistics are different from those of the training dataset, on the prediction of deep learning models has been studied for a long time (Dodge and Karam, 2016; Liu et al., 2021). These DNN models have been known to undergo performance degradation in the presence of such distribution shifts. The model behavior is influenced based on the type of degradation it encounters. In the literature, researchers have worked on exploring the impact of shifts caused due to adversarial noise, label noise, and common corruptions. FR models have been extensively tested against different adversarial attacks due to their key role in security systems. While adversarial noise is crucial for building secure systems, it is associated with the deliberate motive to sabotage model performance.

Another popular line of study involves studying the impact of corruption in *labels*. It refers to the presence of incorrect ground truth labels leading to suboptimal performance of the FR models. Wang et al. (2018a) show how popular training datasets like MegaFace (Kemelmacher-Shlizerman et al., 2016) and MS-Celeb-1M contains over 33% and 67% noisy labels. The authors also show how increasing levels of these noisy labels on a clean dataset from 0% to 50% causes a drop in identification performance of up to 20%. Hu et al. (2019) shows how these noisy labels adversely affect the cosine-similarity of the embeddings of the class centers with the embeddings of the face images in the dataset. In this work, we focus on common corruption where corruption is performed on the images. Dodge and Karam (2016) provided evidence for this degradation in the image classification setting and highlighted the susceptibility of older models, such as VGG-16 and GoogleNet, to blur and noise. Similarly, other older works have highlighted the impact of different degradations on deep FR models.

Common Corruptions refers to the frequently occurring image corruptions such as those of blur, noise, contrast, light, digital transformation, and compression, which change the data distribution by perturbing image properties, keeping the semantic content of the images intact. These common corruptions may get injected into the dataset at any step during image collection, transmission, or generation. From the perspective of FR datasets, they keep the facial features of the images unaffected and easily recognizable by humans. Kara-

han et al. (2016) showed performance degradation under corruptions using the LFW dataset identification protocol. Similarly, Grm et al. (2017) use LFW and VGGFace datasets and observe that high levels of noise, blur, missing pixels, and brightness have a detrimental effect on the verification performance of all models, whereas the impact of contrast changes and compression artifacts is limited. It is important to note that the aforementioned works showcased results using older deep learning models such as AlexNet, VGGFace, and GoogLeNet on the LFW dataset. There is inconsistency in the experimental settings for the different studies, and the face recognition performance of models has since saturated on these easier FR benchmarks (Huang et al., 2008). However, current research in face recognition focuses on datasets with challenging covariates such as age and pose, where the FR models still struggle to generalize (Zheng and Deng, 2018; Sengupta et al., 2016).

Although benchmark datasets like ImageNet-C (Hendrycks and Dietterich, 2019), CIFAR-C (Hendrycks and Dietterich, 2019), Coco-C (Michaelis et al., 2019), etc. exist for evaluating computer vision models on various downstream tasks like classification and object detection in presence of common corruptions, there has been no such benchmark to measure the test efficacy of FR models. This has led to scattered research on the corruption performance of various FR models under different settings, making corruption-based performance comparison between FR models difficult (Grm et al., 2017; Majumdar et al., 2021). Extensive studies in FR literature exist showing the model's performance across various directions like adversarial attacks, age, disguise, morphing, retouching, gender, and other biases (Singh et al., 2020; Cavazos et al., 2020). Another research includes work by Lu et al. (2019) on face verification, studying the impact of multiple covariates such as pose, age, and facial hair on model performance. We believe that in order to study the robustness of models and algorithms in an effective manner, it is imperative to have a common benchmark for comparison. Hence, with the proposed DecordFace framework, we facilitate future research toward the effective study and development of robust FR models in the presence of common image corruptions.

## 3 The DecordFace Framework

The Degraded and Corrupted Face Recognition Evaluation (DecordFace) framework is designed to aid the development of robust face recognition systems through thorough evaluation of a variety of image corruptions. In this section, we describe the key components of the proposed framework and provide details regarding its creation, evaluation protocols, and the proposed metrics. The framework utilizes existing FR datasets (described in Section 4.1) and can be easily expanded for more datasets through the key components described here.

### 3.1 Image Corruptions

As established by previous research, the performance of face recognition is impacted by different covariates, such as lighting and illumination. Through the use of these common degradations and corruptions, we simulate multiple scenarios that may lead to performance deterioration and encourage building systems that are simultaneously robust to these variations. The DecordFace framework consists of 16 image corruptions, namely Gaussian noise, impulse noise, shot noise, speckle noise, defocus blur, Gaussian blur, glass blur, motion

blur, zoom blur, brightness, contrast, saturate, jpeg compression, pixelate, elastic transform, and spatter. Each corruption is applied at five increasing levels of severity. The different corruptions used can be categorized as follows,

- *Blur:* Defocus, Gaussian, Glass, Motion, Zoom

- *Noise:* Gaussian, Impulse, Shot, Speckle

- *Light and Color:* Brightness, Contrast, Saturate

- *Distortion:* Elastic transform

- *Compression:* JPEG compression, Pixelate

- *Occlussion:* Spatter

The sources of these different kinds of corruptions are either during image acquisition or image transmission. The common corruptions in DecordFace are inspired by the ImageNet-C benchmark (Hendrycks and Dietterich, 2019).

The *blur-based variations* lead to a loss in image sharpness where defocus blur occurs when the face of a person is out of focus in the camera, glass blur occurs when a face is viewed through a frosted glass or mirror, motion blur occurs when a face is captured during an abrupt movement of the camera or the person, and zoom blur occurs when the camera or the person move towards each other rapidly. In *noise-based corruptions*, Gaussian noise can occur in low-lighting conditions, while shot noise occurs due to the discrete nature of light, modeled via the addition of random Poisson noise. Impulse noise is caused due to errors in the electronic transmission of images, similar to salt-and-pepper noise. *Light and color-based variations* such as brightness, contrast, and saturation can occur during acquisition due to varying environmental conditions and/or camera properties at the time of capture. *Compression-based corruption* can occur during the transmission and storage of images. Finally, the spatter corruption simulates the *occlusion* of random face parts during image acquisition. Samples from the corruptions can be seen in Fig. 3.

## 3.2 Evaluation Strategy and Protocol

The DecordFace framework utilizes the face verification setting to evaluate different FR models for robustness. In face verification, a pair of face images is utilized and based on the similarity in the representations of faces in the feature space, a decision for their match or non-match is taken. Let a standard FR model $f$ be defined as $z = f(x)$ on a dataset $D$ where $x \in \mathbb{R}^N$ denotes the input image and $z \in \mathbb{R}^d$ denotes the feature embedding of the face image. In the verification setting, the performance is evaluated based on image pairs such as $(x, x')$ with $y$ denoting whether the pair is match or non-match. We use the similarity matcher $\delta$ to define the following function $g$:

$$g(x, x', f, T) = \begin{cases} y_1, & \text{if } \delta(f(x), f(x')) = \delta(z, z') \geq T \\ y_2, & \text{otherwise} \end{cases}$$

where $T$ is the decision threshold, $x' \in \mathbb{R}^N$, $z' \in \mathbb{R}^d$ and $y_1$ and $y_2$ denote the prediction as a match and non-match pair, respectively. $\delta(.,.)$ computes the similarity between the feature embeddings.

---

**Algorithm 1:** Obtaining Corrupted Versions of Datasets for DecordFace

---

**Input:** Face Recognition dataset $\mathcal{D}$ containing $N$ face images ($\{X^t\}_{t=1}^{N}$)

**Initialize:** Initialize the empty corrupted dataset $\mathcal{D}^C$ with 5 severity levels containing 16 corruption types each.

**External Dependencies:** ImageNet-C corruptions library for the common corruptions used. The corruption function $\phi(X^t, s, c)$ takes the original image $X^t$, severity level $s$ and the corruption type $c$ to return the corrupted image $X_{s,c}^t$

**Function** *corrupt($X^t$)*

    **for** *s=1 to 5* **do**

        **for** *c in {defocus_blur, gaussian_blur, glass_blur, motion_blur, zoom_blur, gaussian_noise, impulse_noise, shot_noise, speckle_noise, brightness, contrast, saturate, elastic_transform, jpeg_compression, pixelate, spatter}* **do**

            Corrupt the face image

            $X_{s,c}^t \longleftarrow \phi(X^t, s, c)$

            Save $X_{s,c}^t$ in $\mathcal{D}^C$

        **end**

    **end**

**end**

---

In order to aid the robustness evaluation of the FR models, we utilize one corrupted face image in every verification pair using different corruptions. The set of corruption functions is denoted as $\mathbb{C} = \forall i \{c_i\}$ corresponding to the 16 corruptions shown in Fig. 2. Each corruption is applied at five increasing levels of severity where severity $s \in \mathbb{S} = \{1, 2, 3, 4, 5\}$, leading to a total of *80 corruptions*. The process of obtaining corrupted versions of images is described in Algorithm 1.

Let us assume that $\mathbb{P}_C(c)$ denotes the approximate frequency of common corruptions in the real world. Then, in the standard verification setting, the performance of an FR model is evaluated as $\mathbb{P}_{(x_1,x_2,y) \sim D}(g(x_1, x_2, f, T) = y)$ where on sampling verification pairs $(x_1, x_2)$ from the dataset $D$ with ground-truth $y$, evaluation is performed using the function $g$ defined above. In the DecordFace framework, we propose to evaluate the models under the setting of $\mathbb{E}_{c \sim \mathbb{C}}[\mathbb{P}_{(x_1,x_2,y) \sim D}(g(x_1, c(x_2), f, T) = y)]$, where we explicitly utilize a corrupted image $x_2$ in every verification pair $(x_1, x_2)$.

We study the performance of the FR models as the mean of model performance across all five corruption severities (refer Section 3.1). Additionally, to effectively understand the impact of corruption severity, we propose two evaluation protocols- (i) Low-severity Corruption and (ii) High-severity Corruption Protocol. The first protocol constitutes of model performance at severity $s \in \mathbb{S}_L = \{1, 2, 3\}$, whereas the second protocol focuses on severity $s \in \mathbb{S}_H = \{4, 5\}$. A separate evaluation for low and high corruption severity will allow us to uncover patterns in model performance as the corruption severity varies.

We would like to emphasize that the evaluation for all protocols is conducted using the standard verification pairs for all the datasets. The verification performance for all models is computed using the standard $TPR@FPR$ metric. The FPR values used for the computation of TPR in this framework are reported in Section 4.3. It should be noted

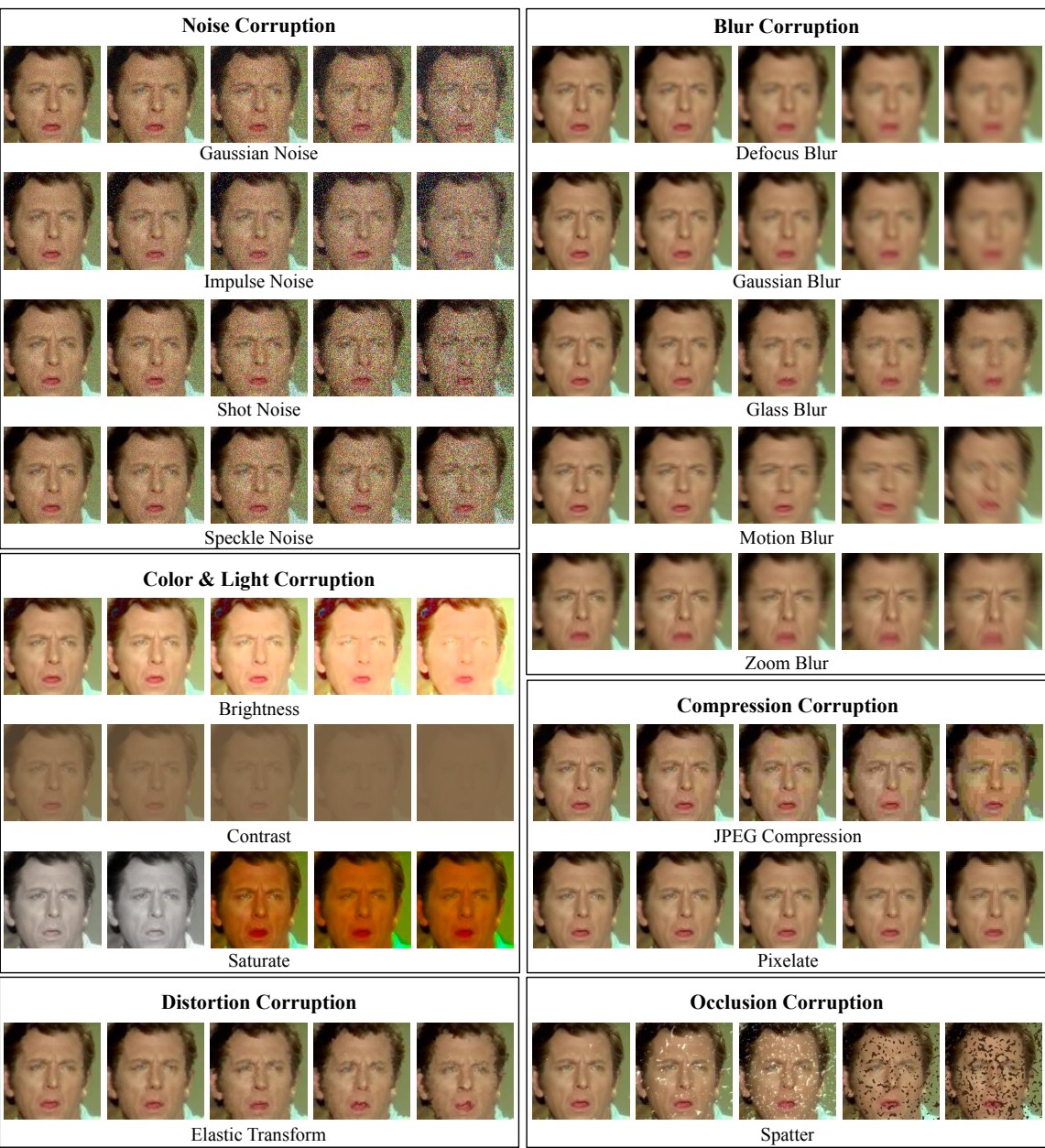

Figure 3: The DecordFace framework contains multiple datasets corrupted with 16 image corruptions. We show all the different corruptions with increasing severity (L-R) on a sample from the AgeDB-*decord* dataset.

that the standard evaluation protocol in FR datasets is largely different from ours in terms of computing the verification performance. We do not conform to the standard evaluation protocol, which performs cross-validation on the smaller datasets to identify the appropriate

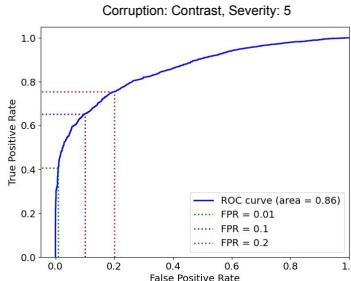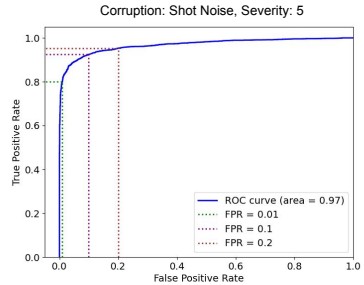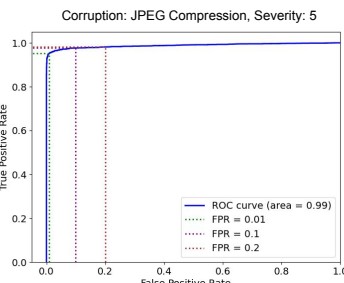

Figure 4: The ROCs showcase the impact of varying FPR thresholds and how they could lead to performance inflation. The performance is shown for three corruptions with disparate impacts on model performance.

decision threshold $T$. This is done in order to avoid using the performance of the model on a subset of data to select the evaluation threshold. We refrain from using any cross-validation-based threshold selection in the design of our protocols and strongly recommend blind evaluation for better estimation during any unseen distribution shift. *Therefore, the DecordFace framework is designed to be used for evaluation only, based on the TPR@FPR protocols specified in the paper.* Our evaluation protocol is in line with the protocols used by the popular large-scale IJB-C dataset.

In our protocol, we employ fixed thresholds rather than cross-validation-based selection, which aligns with established benchmarks such as IJB-C. It ensures that verification accuracy is evaluated at predefined false positive rates, independent of dataset-specific biases. This design better reflects real-world deployment, where thresholds are unlikely to be re-optimized for each new condition. To demonstrate the impact of threshold selection, we provide ROC curves and TPR@FPR analysis for the AgeDB dataset (ArcFace R50, severity level 5) across multiple FPR values (0.01, 0.1, and 0.2) for representative corruptions ((Figure 4)). These analyses clearly show that threshold variations significantly affect model performance, confirming that cross-validation can lead to overly optimistic robustness estimates. While fixed thresholds may penalize models sensitive to threshold placement, they provide fairer cross-model comparisons by applying uniform decision criteria. This ensures that our mVCE metric reflects genuine performance degradation under corruption rather than artifacts of threshold optimization. Similar patterns are observed across other model architectures. These findings support our decision to use fixed thresholds, which provide a fairer and more generalizable robustness evaluation.

### 3.3 Evaluation Metrics and Setup

In this section, we present the various evaluation metrics used to evaluate the corruption robustness of different models. The standard TPR@FPR metric used for evaluating the performance of face recognition models is described first, followed by two new metrics proposed in the DecordFace framework, the mean Verification Corruption Error (mVCE) and the mean Corruption Embedding Invariance (mCEI). The mVCE and mCEI metrics sum-

marize the model performance across different corruptions. While the mVCE metric is specific to the proposed verification protocol, the mCEI metric is computed for the entire corrupted set in conjunction with the clean set.

**TPR@FPR and Error:** The standard TPR@FPR metric calculated for corruption $c$ at severity $s$ is denoted as $TPR_{s,c}^f$, which is computed for a given model $f$. The FPR values are fixed for corruption $c$ at severity $s$. Subsequently, the corresponding verification error is denoted as $E_{s,c}^f$ and computed as,

$$E_{s,c}^f = 100 - TPR_{s,c}^f \tag{1}$$

**Mean Verification Corruption Error (mVCE):** This metric is inspired by the mean Corruption Error metric introduced in ImageNet-C(Hendrycks and Dietterich, 2019). The mVCE metric provides an estimate of the performance degradation of an FR model in the presence of corruptions with respect to the model performance on the non-corrupted face images. For the DecordFace framework, mVCE is computed as the mean verification error for the corruptions over different severities, followed by averaging over all corruptions.

$$VCE_c^f = \frac{1}{|\mathbb{S}|} \sum_{s \in \mathbb{S}} E_{s,c}^f \tag{2}$$

$$mVCE^f = \frac{1}{|\mathbb{C}|} \sum_{c \in \mathbb{C}} VCE_c^f \tag{3}$$

where the verification error $E_{s,c}$ for a corruption $c$ at severity $s$ is computed as detailed in the previous subsection. In other words, the $mVCE$ is computed in three steps. First, the error $E$ is computed from $TPR$ as in Eqn. 1 for a particular corruption $c$ at a specific severity level $s$. After obtaining this error, the $VCE$ of a particular corruption $c$ is computed by aggregating over the 5 severity levels. The $VCE$ value is computed for all corruptions. Finally, to compute the $mVCE$, the $VCE$ values for all the corruptions are averaged.

While the mVCE metric highlights the absolute performance degradation due to the presence of corruptions, it fails to incorporate the possibility where the performance on the non-corrupted (or clean) images is also low. Incorporating the clean verification performance $E_{clean}^f$ allows us to account for the relative drop in model performance due to corruption. With this in mind, we propose another variant of the mVCE metric called the Relative Mean Verification Corruption Error (Relative mVCE) computed using $Relative\ VCE_c^f = (1/|\mathbb{S}|) \sum_{s \in \mathbb{S}} E_{s,c}^f - E_{clean}^f$. It is important to note that while Relative mVCE highlights the relative performance degradation, it is insufficient to measure the robustness of a system by itself. In a scenario where the clean performance of a model is extremely low, a lower Relative mVCE might signal a robust model. Therefore, it is important to evaluate mVCE and Relative mVCE in tandem. The mVCE and Relative mVCE metrics are inversely proportional to the robustness of the model.

**Mean Corruption Embedding Invariance (mCEI):** In face recognition models, it is imperative to understand the impact of covariates on feature embeddings. Since pre-trained models are widely deployed in applications and are essentially used as foundational models in the context of faces, studying the impact of corruptions on the model feature space is

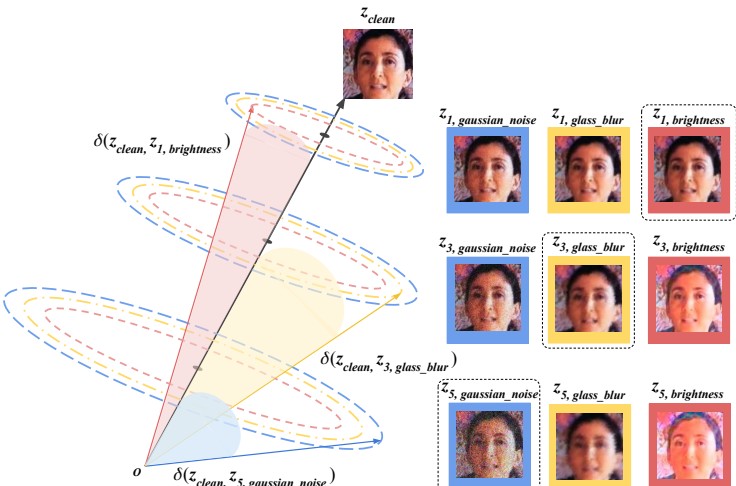

Figure 5: Conceptual representation of the proposed mCEI metric, highlighting the increase in angular distance between the clean and corrupted feature with increasing severity. Here, $z_{clean}$ and $z_{s,c}$ denote the feature embedding of the clean image and image corrupted by corruption $c$ at severity $s$, respectively.

essential. With this goal in mind, we propose the Mean Corruption Embedding Invariance (mCEI) metric. This metric functions outside of the verification protocol but evaluates the shift in model embeddings from clean to corrupt faces. We conceptually illustrate the mCEI metric in Fig. 5, where the core idea is to capture the angular distance between the clean and corrupted feature embedding, taking into account the type and severity of corruption. We compute the mCEI metric first for each corruption and then for all severities together as follows,

$$CEI_c^f = \frac{1}{|\mathbb{S}|} \sum_{s \in \mathbb{S}} \delta \left( z_{s,c}^f, z_{clean}^f \right) \tag{4}$$

$$mCEI^f = \frac{1}{|\mathbb{C}|} \sum_{c \in \mathbb{C}} CEI_c^f \tag{5}$$

where $\delta(.,.)$ computes the cosine similarity between the feature embeddings. For computing $CEI$ for a corruption $c$, the similarity is first calculated at a particular severity $s$ with the clean embedding. Then, this process is repeated for every severity, and the similarity values are then averaged across the severities. Finally, to compute $mCEI$, the $CEI$ values are aggregated across all the corruptions.

Measuring the cosine angle between the original and corrupt embeddings of the corresponding clean and corrupt input image provides insight into the shift caused in model embeddings due to corruption. The mCEI metric provides perfect performance when the model embeddings corresponding to the clean and corrupted images are completely invariant. It is directly proportional to model robustness. This metric can be utilized to study the effect of any and all variations in an input image, where the feature is expected to be

Table 1: Summarizing the details about the number of images and verification pairs available in the datasets and the corresponding corrupted versions generated for the DecordFace framework. A total of 80 variations (16 corruptions × 5 severity levels) of the images and pairs are generated for DecordFace. Since the IJB-C dataset uses template- template verification protocol, all 469k images were corrupted first. The number of template images and pairs is reported. *(datasets cited in text)*

| Dataset Name | Test Images (Original) | Test Pairs (Original) | Images (DecordFace) | Pairs (DecordFace) |
|---|---|---|---|---|
| AgeDB | 5298 | 6000 | 423k | 480k |
| CALFW | 7156 | 6000 | 572k | 480k |
| CFP-FP | 5901 | 7000 | 472k | 560k |
| CPLFW | 5984 | 6000 | 478k | 480k |
| IJB-C | 23k | 15.6M | 1.8M | 125M |

invariant. In addition to image-based corruptions, the fundamental rationale behind the $CEI$ metric may be utilized for studying adversarially attacked images. Further, this metric may be utilized in the design of robust algorithms.

The mVCE and mCEI metrics provide separate views of the influence of corruption on the model performance. Under our protocol, verification decisions depend on whether similarity scores exceed a fixed threshold. Given that the models are fixed and trained only on clean data, a high similarity between clean and corrupted embeddings leads to a strong correlation: a higher mCEI implies that distortions cause minimal embedding drift, leading to lower error rates, while a lower mCEI aligns with degraded performance. However, there are certain cases where this relationship may falter. For example, if the models are poorly trained for face recognition. In that scenario, if clean embeddings are already poorly separated, even a high mCEI may not translate to correct classification because the decision boundaries are unreliable. Further, mCEI is a dataset-level metric. The verification performance is dependent on the specific genuine and impostor pairs. Since mCEI averages over samples, it may remain high if most embeddings are stable, even if a few borderline samples cross the threshold and cause errors. If the verification set consists largely of these error cases, mCEI would not be a reliable indicator of model performance. While a high mCEI is likely to indicate a low mVCE, the same may not be true vice versa. Therefore, we believe that it is important to compute, analyze, and report both these metrics for a better understanding of the underlying models in the presence of corruptions.

## 4 Experimental Design

In this section, we discuss the different datasets that form the DecordFace framework. Next, we discuss the recognition models used for performing the baseline experiments. Finally, the details of implementation are provided.

### 4.1 Datasets

For the framework, we apply the corruptions to the test images of five standard FR datasets, namely AgeDB (Moschoglou et al., 2017), CALFW (Zheng et al., 2017), CFP-FP (Sengupta

et al., 2016), CPLFW (Zheng and Deng, 2018), and IJB-C (Maze et al., 2018) to create their corresponding corrupted variants, namely AgeDB-*decord*, CALFW-*decord*, CFP-FP-*decord*, CPLFW-*decord*, and IJB-C-*decord*. The AgeDB and CALFW are popular frameworks for capturing age variation, whereas CFP-FP and CPLFW contain high pose variation. The IJB-C dataset is a large-scale face dataset containing multiple covariates with over 15M test pairs, allowing for the evaluation of models at lower FPRs. These datasets contain variations across age, pose, and overall face image quality. The dataset statistics are summarized in Table 1, and further details are provided below. Collectively, the DecordFace framework comprises over 126 million image verification pairs. We provide details of the datasets below.

**AgeDB (Moschoglou et al., 2017):** The AgeDB-*decord* is derived from the standard AgeDB dataset, which is widely used for evaluating the performance of a face recognition (FR) model. The images in the verification pairs differ by a significant age difference. We use the 30-year age gap protocol widely known as the AgeDB30 used in standard evaluation protocols. AgeDB30 contains 6000 verification pairs, out of which 3000 are genuine pairs and the remaining are impostor pairs. We apply a total of 80 corruptions (refer Section 3.1) on the 5298 images present in the dataset and obtain approximately 423,840 corrupted images. Using the 6000 test pairs provided in the dataset, we construct variants across the 80 corruptions (as explained in Section 3.2) leading to 480k (6000 x 80) evaluation pairs.

**CALFW (Zheng et al., 2017):** Similar to the AgeDB dataset, the CALFW dataset contains faces with age variation and is widely used for evaluating FR models. The dataset images consist of an age difference between 5 to 27 years. The standard test protocol contains a total of 6000 verification pairs, out of which 3000 pairs are genuine, and the remaining are impostor pairs. The images and pairs in DecordFace are created through the corruptions and are obtained to be approximately 572k (7156 x 80) and 480k (6000 x 80), respectively.

**CFP-FP (Sengupta et al., 2016):** CFP-FP-*decord* is derived from the standard CFP-FP dataset, which is widely used for evaluating FR models. The images in the verification pair have significant pose differences, specifically frontal and profile views. The original dataset consists of two protocols- frontal-frontal and frontal-profile. The frontal-profile protocol is adopted for evaluation; therefore, we create CFP-FP-*decord* corresponding to the second protocol. The CFP-FP dataset contains 7000 verification pairs, out of which half are genuine pairs and the remaining half are impostor pairs. Then, CFP-FP-*decord* contains approximately 472k (5901 x 80) images and 560k (7000 x 80) evaluation pairs.

**CPLFW (Zheng and Deng, 2018):** CPLFW-*decord* is created using the CPLFW dataset. The dataset contains high pose variation. We utilize the images corresponding to the standard verification protocol for creating CPLFW-*decord*. The CPLFW dataset contains 6000 verification pairs, out of which 3000 are genuine pairs and the remaining are impostor pairs. As a result, CPLFW-*decord* contains 478k (5984 x 80) images and 480k (6000 x 80) evaluation pairs.

**IJB-C (Maze et al., 2018):** For the IJB-C-*decord* dataset, we use the IJB-C dataset. Specifically, we use the 1:1 verification test protocol. This protocol contains over 15M verification pairs created using 23k images in the test set. The testing is performed through templates created from images present in the IJB-C dataset. As per the standard protocol, there are 19.5k matching template pairs, whereas the remaining are non-matching. Using

the 23k test images, we create 1.8M images (23,000 x 80 = 1,840,000). Similarly, corresponding corrupted image pairs are created for the 15.6M test pairs, leading to a total of 125M evaluation pairs.

## 4.2 Face Recognition (FR) Models

We consider several FR algorithms for evaluation of the DecordFace framework, including LightCNN (Wu et al., 2018), Arcface (Deng et al., 2022), Cosface (Wang et al., 2018b), MagFace (Meng et al., 2021b), ElasticFace (Boutros et al., 2022), Adaface (Kim et al., 2022), and Controllable Face Synthesis Model (CFSM) (Liu et al., 2022), leading to a total of 27 pre-trained FR models. For each algorithm, we consider multiple backbones. For LightCNN, we use models with the 9-layer, 29-layer, and 29-layer v2 backbones. For Cosface and ArcFace, we use models with ResNet18, ResNet34, ResNet50, and ResNet100 backbones. For AdaFace, we use models with iResNet18, iResNet34, iResNet50 and iResNet100 backbones pre-trained on three different datasets. These models are selected based on their popularity as well as their recency and efficiency in FR literature. The details of the models, as well as their publicly available repositories, are presented here. A summary of the models used can also be seen through Columns 1-3 of Table 2 presented as part of the results.

**LightCNN (Wu et al., 2018)**[2]**:** is a light-weight model which learns a compact embedding of the face image. In this model, the authors introduced the Max-Feature-Map (MFM) module, which separates noisy and informative signals from the image as well as performs effective feature selection.

**Cosface (Wang et al., 2018b)**[3]**:** uses an angular margin loss termed as large margin cosine loss (LMCL) which replaces the standard softmax loss. The authors show that using the LMCL loss maximizes the decision margin in the angular space, leading to improved recognition performance.

**Arcface (Deng et al., 2022)**[4]**:** has arguably been the most widely used loss function in recent years for face recognition. The ArcFace model uses an angular margin loss to enhance the discriminative power of the network and has been shown to improve FR performance considerably.

**MagFace (Meng et al., 2021b)**[5]**:** is a relatively newer FR recognition algorithm that utilizes a loss function that enables the learning of a universal feature embedding such that it is aware of the quality of a given face image. This is achieved through an adaptive learning mechanism that pushes easy samples toward class centers while pushing noisy samples away to prevent overfitting.

**ElasticFace (Boutros et al., 2022)**[6]**:** uses a loss function that builds on top of margin losses such as CosFace and ArcFace by updating the fixed penalty margin. The penalty margin is updated with a variable component, allowing for more flexibility in the learning of the margin.

---

2. `https://github.com/AlfredXiangWu/LightCNN`

3. `https://github.com/deepinsight/insightface`

4. `https://github.com/deepinsight/insightface`

5. `https://github.com/IrvingMeng/MagFace`

6. `https://github.com/fdbtrs/ElasticFace`

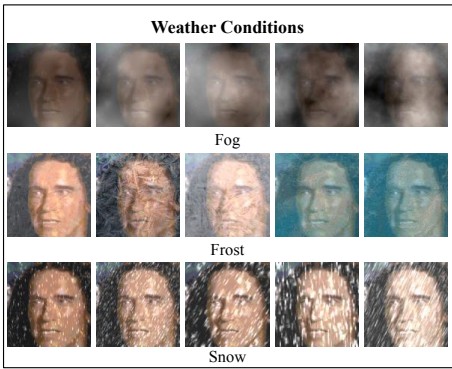

Figure 6: Samples showcasing the unrealistic appearance of weather-based corruptions on face images. Modeling realistic weather conditions on faces, while out of scope for our work, remains an interesting future avenue for research.

**Adaface (Kim et al., 2022)[7]:** is a recent FR model that utilizes an adaptive margin function that factors the quality of the image sample while training. This leads to a quality-aware FR model and has been shown to provide high-performance improvements over previous models.

**CFSM (Liu et al., 2022)[8]:** is another recent FR model that introduces a face synthesis model for face recognition such that learns the style of the target distribution during training. As a result, it is able to learn better face representations. The authors have shown promising results on face recognition using CFSM-based models.

### 4.3 Implementation Details

In this section, we discuss the implementation details involved in the creation of the DecordFace framework as well as its evaluation.

**Dataset Pre-processing:** The datasets included in DecordFace are derivatives of popular benchmark datasets. Since most of the datasets are provided unaligned and uncropped, we first crop and align the required images from the original datasets using the RetinaFace detection algorithm (Deng et al., 2020) (`https://github.com/hukkelas/DSFD-Pytorch-Inference`). In cases where there were multiple faces detected in the image, the cropped face with the maximum score, which was closest to the center of the image, was retained. This was further verified manually to ensure the correspondence between the cropped face and the reported identity. The detected 5-point facial landmarks are used to align the detected faces using affine transform, with the reference points being the 5 landmarks standardized by ArcFace. The images are cropped to $112 \times 112$ for all the models.

**Image Corruption:** After face detection, we perform the image corruption. Face detection is performed first to ensure that the verification performance or verification pairs are not

---

7. `https://github.com/mk-minchul/AdaFace`
8. `https://github.com/liufeng2915/CFSM`

impacted by the limitations of the face detection algorithm used. Face images are corrupted with the 16 corruption types and 5 level of severities. We use the same corruption pipeline as used by ImageNet-C (`https://github.com/hendrycks/robustness`). We modified the code for elastic transform as it appeared to provide incorrect transformations on the images. We do not include snow, frost, and rain corruption because of their unrealistic appearance. Some samples are shown in Figure 6. For faces captured close-up (e.g., ID verification, mugshots, many surveillance scenarios), these corruptions may not realistically appear on the face as demonstrated by the sample images. We acknowledge that modeling authentic weather-based corruptions for long-range outdoor surveillance represents a valuable direction for future research. In this work, 80 variations of the datasets corresponding to all the $5 \times 16$ severity and corruption combinations are created.

**Feature Extraction and Evaluation:** After the corrupted datasets have been created, feature extraction is performed using the 27 selected pre-trained models. The input image transformations for each of the models are applied as per their respective GitHub repositories. The extracted features for the clean and corrupted images are used for computing TPR@FPR as per the protocol and subsequently for computing the mVCE and mCEI metrics. For AgeDB-*decord*, CALFW-*decord*, CPLFW-*decord*, and CFP-FP-*decord*, $TPR$ is calculated at $FPR = 1e-2$. For the IJB-C dataset, we employ the 1:1 verification protocol using image templates. The template $x_2$ is constructed using corrupted images, and the results are computed at FPRs of $1e-6$, $1e-5$, and $1e-4$. The severities set $\mathbb{S}$ used for computing the mVCE and mCEI metrics is considered under three settings- $\mathbb{S}_H = \{4,5\}$, $\mathbb{S}_L = \{1,2,3\}$, and $\mathbb{S} = \{1,2,3,4,5\}$ for high severity, low severity, and overall severity, respectively. The similarity between the feature embeddings $\delta$ utilizes cosine similarity as the similarity measure.

**Computational Resources:** In this section, we elaborate on the computational resources and approximate time taken for the steps involved in the creation of the DecordFace framework, as well as for computing results using 27 FR models on it. The RetinaFace detector was used for face detection with a batch size of 64, and required approximately 5 minutes to detect, crop, and align all the faces from the images in each of the smaller datasets (namely AgeDB, CPLFW, CALFW, and CFP-FP) on an NVIDIA RTX 3090 GPU machine with 24 GB VRAM. IJB-C, being a considerably larger dataset, took approximately 8 hours to finish detection and alignment on the same machine. Using the same machine, with Multi-Processing on a 48 core AMD Threadripper CPU, all the images of the smaller datasets were corrupted in approximately 25 minutes each. The IJB-C took approximately three days to corrupt all the images. These corrupted images form our DecordFace dataset. The most compute-intensive part of the paper was evaluating a large number of models with various backbones on all the DecordFace framework datasets. This part of the workflow was executed parallelly on an NVIDIA DGX Station with 3 NVIDIA V100 GPU with 32 GB each and on a local machine with NVIDIA RTX 3090 GPU. On IJB-C-*decord*, each of the R18 and R34 backbones took around 6 hours, whereas the R50 and R100 backbones took around 8 and 10 hours, respectively. All the FR models on the entire IJB-C-*decord* dataset finished computing in about 10 days. The feature extraction for the smaller datasets using all the models finished in 8 hours each. The evaluation was enhanced using multi-processing. Finally, computing the verification scores for the mVCE metric and the similarity scores for the mCEI metric on the IJB-C dataset required 24 hours due to the large number of im-

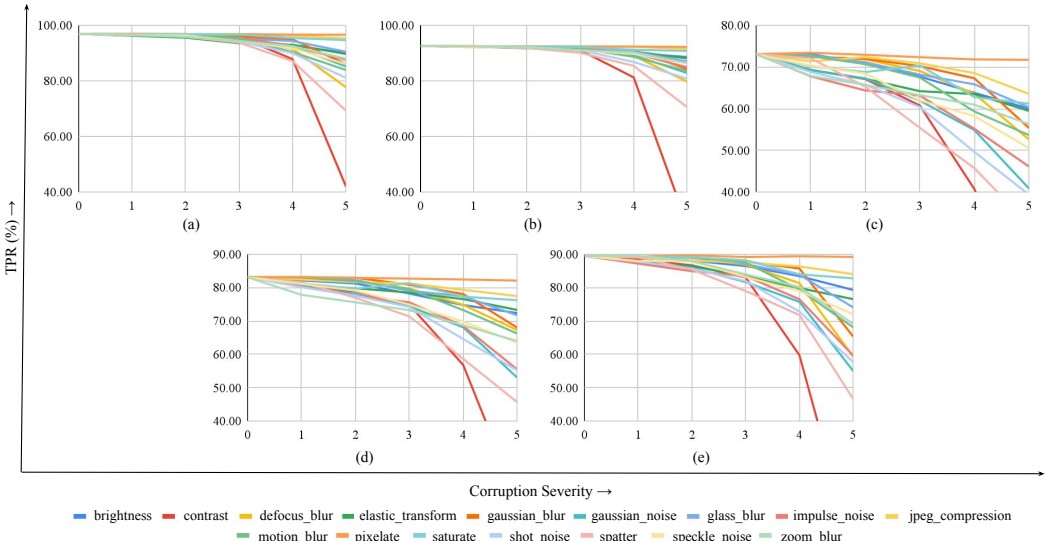

Figure 7: Trend of drop in TPR@FPR with increasing severity of corruptions in (a) AgeDB-*decord* (b) CALFW-*decord* (c) CPLFW-*decord* (d) CFP-FP-*decord* (e) IJB-C-*decord* datasets. The results are reported for the ArcFace model with ResNet50 backbone. The TPR decreases with increasing severity across all data subsets.

ages and verification pairs. All experiments are conducted on LINUX-based systems using Python-based libraries, and specifically, the PyTorch library is used for all experiments.

## 5 Results and Analysis

In this section, we analyze the performance of different models as impacted by the corrupted input data provided to them. Intuitively, the model performance should worsen as the severity of the corruption increases. This phenomenon is observed for different datasets utilized in the DecordFace framework across multiple models and is depicted in Fig. 7 for the ArcFace model with R50 backbone. While datasets with age variation, such as AgeDB-*decord* and CALFW-*decord*, are strongly impacted at higher severities, other datasets are observed to be impacted at even lower severities. This drop in model performance suggests a lack of robustness in models at the presentation of these common corruptions. The results obtained on the DecordFace framework are presented below. We begin with an analysis based on the *TPR* performance of the models across the different corruptions and severities. Then, we discuss the model performance based on the proposed mVCE and mCEI metrics. Finally, we analyze the impact of model architecture and training methods on model performance, followed by a brief fairness-based analysis.

In order to understand the impact of different corruptions across the models, we analyze the performance drop in different models through Fig. 8. The results are provided for the IJB-C-*decord* dataset computed at FAR 1e-4. The performance drop in TPR of each model is binned based on four bins. Each bar visually depicts the number of models falling into

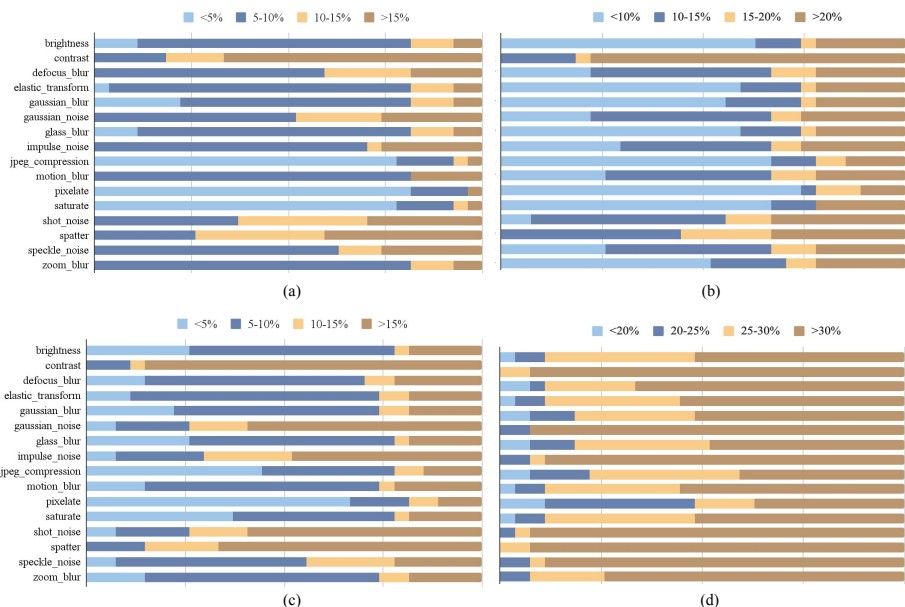

Figure 8: Plot highlighting the relationship between the evaluated models and their % drop in TPR due to different corruptions for the (a) AgeDB-*decord* (b) CALFW-*decord* (c) IJB-C-*decord* and (d) CFP-FP-*decord* datasets. The results are reported over all severities. Sharp performance drops are observed due to contrast, while compression-based corruptions have a low impact.

a bin corresponding to a corruption. From the plot, we identify that compression-based corruptions such as jpeg compression and pixelated (refer to compression corruptions in Fig. 3) have a low impact on model performance as most models fall into the lowest performance drop bin. This behavior is consistent with observations made in previous literature (Grm et al., 2017). However, we observe sharp performance drops due to contrast, which has been shown not to impact model performance (Grm et al., 2017). We also observe high drops in the range of 5-10% for other light and color-based corruptions, such as brightness and saturation for most datasets. The corruptions that impact the model performance the most constitute contrast, Gaussian noise, impulse noise, and shot noise. Similarly, a large number of models are impacted with a 5-10% performance drop for blur-based corruptions. These corruptions can easily occur due to poor lighting conditions, bit errors, etc. (refer to blur corruptions in Fig. 3), and therefore, it is important for models to be robust against these corruptions. Another corruption that is detrimental to model performance, causing more than 15% performance drop, is spatter, which occludes random parts of the face image. Even when the % drops are relatively higher for datasets such as CPLFW, the overall impact of a given corruption with respect to the other corruptions on the dataset remains the same.

At high severity, contrast and noise-based corruptions are detrimental to model performance, followed by blur-based and light and color-based corruptions. We believe that these

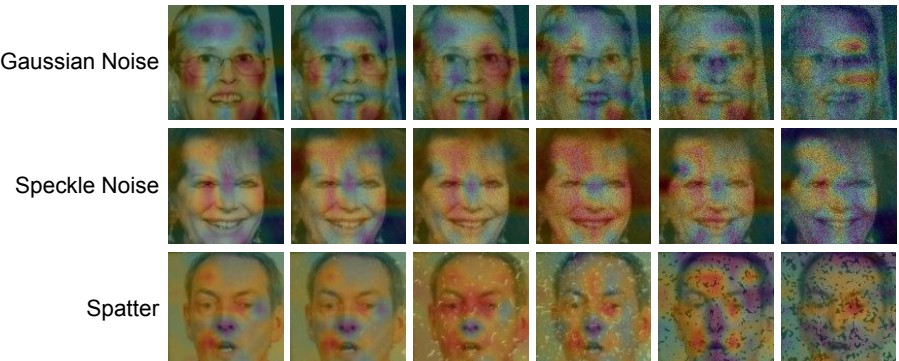

Figure 9: Activation Maps on the DecordFace framework highlighting shift in model's regions-of-interest on prediction using ArcFace R100. The first sample shown corresponding to each corruption is clean, followed by increasing levels of severity for each corruption (L-R). For all the samples presented here, the model provides correct verification prediction using the clean sample but misclassifies when presented with the corrupted sample. The clean images belong to the AgeDB, CALFW, and CPLFW datasets.

corruptions cause a shift towards less discriminative regions of interest in the model, leading to poorer performance. To further understand the change in model behavior, we employ activation maps to visualize the shift in regions of interest in the presence of corruptions. In Fig. 9, we provide multiple such samples which are corrupted using glass blur, spatter, speckle noise, and zoom blur. It can be observed that the model's focus shifts from discriminative regions such as the eyes, nose, and mouth towards less discriminative features such as the forehead, cheek, and chin. This explains the misclassification of the samples.

## 5.1 Quantifying mVCE Corruption Performance

In this section, we report the performance of different models when evaluated using the proposed mVCE metric. Table 2 shows the performance for this metric for the high severity protocol. The results for low severity and overall severity are reported in Tables 8 and 9. From the table, it can be observed that the mVCE values differ significantly for different evaluation datasets. This behavior is expected as different datasets present varying degrees of difficulty for verification. Based on the trends observed in Table 2, the CPLFW-*decord* and CFP-FP-*decord* datasets are impacted the most in their verification performance. This trend presents across the different model backbones, indicating the high difficulty in identifying pose-based variation. A similar trend is observed in the Relative mVCE for CPLFW-*decord* and CFP-FP-*decord* datasets for high severity, indicating that models tend to fail more in the presence of pose variation along with corruptions. The results for the Relative mVCE are provided in Tables 10, 11, and 12, corresponding to the high severity protocol, low severity protocol, and overall performance, respectively.

Table 2: The mean Verification Corruption Error (mVCE) performance (%) of the different models on the DecordFace framework for the *high severity* corruption protocol. The higher the mVCE, the less robust the model is towards corruptions. 'R' and 'iR' in column 2 refer to ResNet and iResNet backbones, respectively. * denotes error on non-corrupted data greater than 40%. *(models cited in text)*

| Model Name | Backbone | Pretraining Dataset | AgeDB -decord | CALFW -decord | CPLFW -decord | CFP-FP -decord | IJB-C-*decord* | | |
|---|---|---|---|---|---|---|---|---|---|
| | | | | | | | 1e-4 | 1e-5 | 1e-6 |
| LightCNN | 9L | MS-Celeb-1M + CASIA-WebFace | 60.40 | 57.09 | 82.81* | 65.14 | 60.67 | 71.63 | 81.71 |
| LightCNN | 29L | | 43.54 | 42.84 | 72.97* | 47.54 | 43.17 | 58.47 | 71.57 |
| LightCNN | 29Lv2 | | 37.55 | 39.37 | 69.85* | 43.16 | 40.56 | 63.17 | 82.66 |
| CosFace | R18 | Glint360k | 27.74 | 25.19 | 64.65 | 45.46 | 24.63 | 34.25 | 44.05 |
| CosFace | R34 | Glint360k | 14.24 | 16.30 | 46.94 | 30.33 | 14.11 | 22.58 | 33.46 |
| CosFace | R50 | Glint360k | 10.60 | 13.74 | 39.00 | 23.03 | 12.31 | 23.73 | 41.84 |
| CosFace | R100 | Glint360k | 9.28 | 12.84 | 34.65 | 18.80 | 12.69 | 25.16 | 50.87 |
| ArcFace | R18 | MS1MV3 | 27.61 | 26.22 | 64.72 | 51.87 | 22.72 | 31.57 | 41.12 |
| ArcFace | R34 | MS1MV3 | 16.62 | 18.04 | 51.73 | 38.99 | 16.56 | 24.21 | 35.72 |
| ArcFace | R50 | MS1MV3 | 11.80 | 14.58 | 44.92 | 32.59 | 13.39 | 19.51 | 27.85 |
| ArcFace | R100 | MS1MV3 | 9.53 | 13.39 | 38.93 | 27.76 | 11.94 | 17.90 | 28.65 |
| MagFace | iR100 | MS1MV3 | 13.60 | 15.78 | 43.71 | 29.51 | 20.13 | 28.83 | 39.13 |
| ElasticFace-Arc | iR100 | MS1MV2 | 13.63 | 16.98 | 43.53 | 30.00 | 18.61 | 26.96 | 39.28 |
| ElasticFace-Cos | iR100 | MS1MV2 | 14.69 | 16.33 | 42.75 | 28.54 | 16.91 | 25.73 | 38.98 |
| ElasticFace-Arc+ | iR100 | MS1MV2 | 14.02 | 16.27 | 42.17 | 28.43 | 16.86 | 24.14 | 35.43 |
| ElasticFace-Cos+ | iR100 | MS1MV2 | 15.84 | 17.02 | 42.66 | 29.49 | 17.56 | 26.17 | 39.42 |
| AdaFace | iR18 | VGGFace2 | 48.01 | 40.28 | 65.12* | 46.44 | 33.00 | 45.97 | 57.41 |
| AdaFace | iR18 | CASIA-WebFace | 54.34 | 52.05 | 98.62* | 65.00 | 97.78* | 99.89* | 99.99* |
| AdaFace | iR50 | CASIA-WebFace | 40.76 | 40.56 | 88.67* | 47.03 | 96.22* | 99.92* | 99.98* |
| AdaFace | iR50 | MSIMV2 | 13.51 | 14.10 | 44.63 | 28.99 | 21.82 | 33.13 | 45.91 |
| AdaFace | iR100 | MS1MV2 | 10.08 | 12.48 | 37.00 | 23.99 | 16.09 | 25.40 | 38.01 |
| AdaFace | iR100 | MS1MV3 | 8.60 | 10.91 | 35.67 | 24.42 | 9.04 | 14.84 | 23.59 |
| AdaFace | iR18 | WebFace4M | 31.35 | 25.73 | 58.43 | 41.38 | 20.41 | 31.17 | 42.56 |
| AdaFace | iR50 | WebFace4M | 12.89 | 13.90 | 35.02 | 21.28 | 7.50 | 12.96 | 21.93 |
| AdaFace | iR100 | WebFace4M | 9.19 | 11.13 | 26.96 | 13.95 | 5.56 | 9.43 | 16.21 |
| AdaFace | iR100 | WebFace12M | 7.59 | 10.10 | 25.13 | 13.40 | 4.92 | 9.07 | 21.95 |
| CFSM-Arc | iR50 | Cleaned MS1MV2 | 26.65 | 15.74 | 43.47 | 30.87 | 14.70 | 21.10 | 29.23 |

Table 3: The mean Corruption Embedding Invariance (mCEI) performance (%) for the different models on the *high severity* corruption protocol. Higher values of mCEI indicate high invariance and greater robustness. *(models cited in text)*

| Model Name | Backbone | Pretraining Dataset | AgeDB -decord | CALFW -decord | CPLFW -decord | CFP-FP -decord | IJB-C -decord |
|---|---|---|---|---|---|---|---|
| LightCNN | 9L | MS-Celeb-1M + CASIA-WebFace | 56.75 | 56.78 | 55.99 | 55.67 | 59.90 |
| LightCNN | 29L | | 66.58 | 65.87 | 63.58 | 62.83 | 69.50 |
| LightCNN | 29Lv2 | | 67.91 | 67.49 | 64.18 | 65.04 | 71.09 |
| CosFace | R18 | Glint360k | 63.34 | 65.56 | 63.19 | 63.31 | 70.66 |
| CosFace | R34 | Glint360k | 69.16 | 71.11 | 67.59 | 68.01 | 76.20 |
| CosFace | R50 | Glint360k | 71.28 | 73.08 | 70.12 | 70.20 | 78.05 |
| CosFace | R100 | Glint360k | 71.92 | 74.36 | 71.58 | 71.52 | 78.08 |
| ArcFace | R18 | MS1MV3 | 64.51 | 64.15 | 61.15 | 60.83 | 69.27 |
| ArcFace | R34 | MS1MV3 | 68.11 | 68.59 | 63.84 | 64.49 | 72.85 |
| ArcFace | R50 | MS1MV3 | 70.30 | 70.68 | 66.19 | 66.20 | 75.01 |
| ArcFace | R100 | MS1MV3 | 71.68 | 71.84 | 66.83 | 67.48 | 76.42 |
| MagFace | iR100 | MS1MV3 | 70.37 | 70.55 | 65.49 | 66.85 | 74.07 |
| ElasticFace-Arc | iR100 | MS1MV2 | 67.55 | 67.73 | 61.83 | 63.59 | 71.07 |
| ElasticFace-Cos | iR100 | MS1MV2 | 65.72 | 65.50 | 61.19 | 62.09 | 69.86 |
| ElasticFace-Arc+ | iR100 | MS1MV2 | 67.67 | 67.79 | 62.51 | 63.89 | 71.74 |
| ElasticFace-Cos+ | iR100 | MS1MV2 | 64.85 | 64.55 | 60.81 | 61.55 | 69.59 |
| AdaFace | iR18 | VGGFace2 | 65.94 | 64.83 | 60.12 | 61.34 | 67.94 |
| AdaFace | iR18 | CASIA-WebFace | 62.01 | 60.06 | 56.61 | 58.69 | 64.55 |
| AdaFace | iR50 | CASIA-WebFace | 67.76 | 66.43 | 61.99 | 64.48 | 70.91 |
| AdaFace | iR50 | MSIMV2 | 70.33 | 69.63 | 65.03 | 65.88 | 72.66 |
| AdaFace | iR100 | MS1MV2 | 71.40 | 70.53 | 65.91 | 66.98 | 73.74 |
| AdaFace | iR100 | MS1MV3 | 73.34 | 72.98 | 68.98 | 69.01 | 77.39 |
| AdaFace | iR18 | WebFace4M | 69.73 | 68.30 | 66.71 | 65.81 | 72.24 |
| AdaFace | iR50 | WebFace4M | 75.53 | 74.83 | 73.33 | 71.74 | 80.56 |
| AdaFace | iR100 | WebFace4M | 77.22 | 76.56 | 75.01 | 73.65 | 82.64 |
| AdaFace | iR100 | WebFace12M | 76.69 | 75.80 | 74.09 | 72.72 | 81.86 |
| CFSM-Arc | iR50 | Cleaned MS1MV2 | 61.38 | 70.88 | 65.52 | 66.70 | 74.67 |

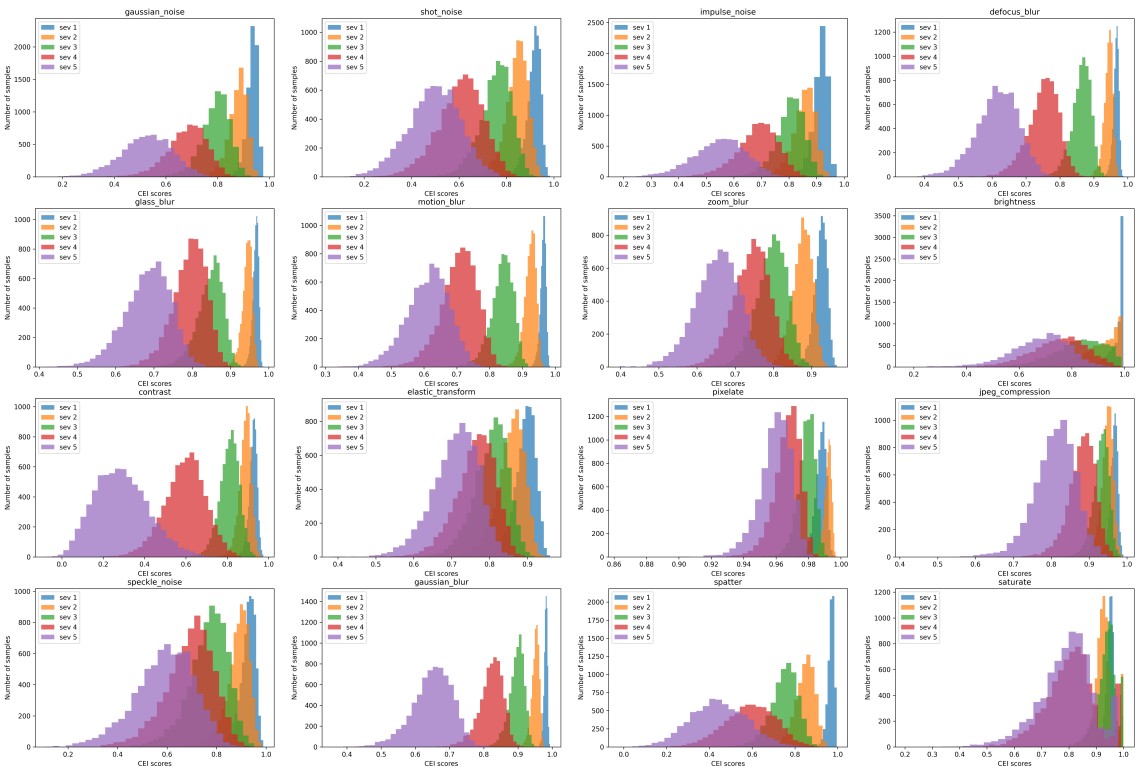

Figure 10: The distribution of CEI scores for the CALFW dataset obtained using the feature embeddings of the Adaface R18 model. The distribution across the five severities highlights the impact of a given corruption on the feature embeddings.

A comparison between the mVCE and Relative mVCE at low severity highlights that an overall improvement in FR performance could automatically improve the robustness of the models towards mVCE. However, at high severity, we can observe that the error values are very high, even for Relative mVCE. While the models with deeper backbones provide reasonably low mVCE values at low severity, indicating their robustness (Table 8). It is important to note that shallower models such as ResNet18 and ResNet50 backbones lead to high mVCE values with upto 26.99% mVCE on CFP-FP-*decord* dataset. This shows that there is a huge scope for improving model robustness, specifically for shallower model backbones.

**Statistical Significance of Drop in Verification Performance:** To understand and validate whether the error generated by the models varies significantly across the five severity levels for a given corruption, we perform significance testing using the paired t-test. The test is performed between every two consecutive transitions for a given corruption from severity 1 to severity 5. We test the null hypothesis that the means of the distributions underlying the verification scores are equal. At a confidence level of 0.95, we observe p-values $< 0.05$ for the majority of the corruption severities in the DecordFace datasets, indicating

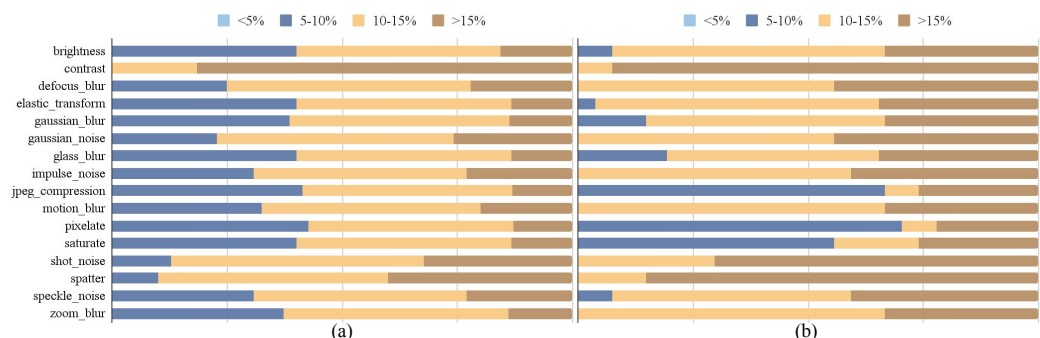

Figure 11: Plot highlighting the relationship between the evaluated models and their % drop in TPR due to different corruptions on (a) male and (b) female subgroups. The female subgroup is impacted by noise significantly more than the male subgroup.

that difference between the underlying distributions and the corruptions affect the model performances significantly from one severity level to the next. However, under the pixelate corruptions from severity 1 to 2, models such as LightCNN and certain variants of AdaFace accept the null hypothesis, depicting the performance difference to be insignificant (on AgeDB-*decord* and CFP-FP-*decord* datasets). The same observations can be made for the saturation corruption from severity 2 to 3, indicating that the models are not impacted much differently by these corruptions (on AgeDB-*decord* and CALFW-*decord* datasets). However, these cases are limited. Similarly, on performing paired t-test between the distribution of scores obtained on non-corrupted images and distribution obtained after using face images with different severity levels, we observe at a confidence level of 0.95, p-values $< 0.05$ for the majority of the corruption severities in the DecordFace datasets, indicating that there is a difference between the two distributions is significant with the exception of the pixelate and saturate corruptions for the AgeDB and CFP-FP datasets. This is consistent with the observations made through the model's mVCE scores for the datasets.

## 5.2 Quantifying mCEI Corruption Performance

In this section, we evaluate the performance of different models under corruption using the mCEI metric. As explained in Section 3.3, the goal of designing this metric is to understand the influence of corruptions on model-generated face embeddings. The results obtained for this metric at high severity are shown in Table 3, and for low severity and overall severity are shown in Tables 13 and 14, respectively. Intuitively, there should be an inverse correlation between a model's mVCE and mCEI performance since model embeddings are utilized for computing verification performance. This behavior is observed across the board where severely impacted CPLFW-*decord* and CFP-FP-*decord* datasets present with lower mCEI values. In an ideal scenario, the embedding containing facial identity features should not be influenced by corruption variations. It is interesting to observe that at high severity, all models report an mCEI value lesser than 83%, showing that the face embeddings changed by at least 17% in the presence of corruptions. On the other hand, at low severity, the

mCEI values range from 84-93%, showing relatively higher invariance to corruptions. This trend in mCEI values can be clearly observed in Fig. 10. The mCEI values are consistently higher for the IJB-C dataset. We believe this occurs as a result of a reduction in corruption noise due to the averaging of multiple images in a template. Except for IJB-C, the mCEI values for a given model are observed to vary within a tight range across different datasets, highlighting consistent model behavior in the presence of corruptions. It should be noted that a high mCEI does not necessarily indicate high model performance since a model with poor clean embeddings might report a high mCEI with equally poor corrupt embeddings. Hence, we recommend reporting both mVCE and mCEI metrics for proper evaluation on the DecordFace framework.

### 5.3 Visualizing Feature Space

To complement the mCEI metric, we use dimensionality reduction and t-SNE to visualize how embeddings of clean and corrupted images shift in latent space. The t-SNE is shown in Figure 12. The t-SNE visualizations were generated using 200 random samples from the AgeDB dataset corrupted at severity level 5, with features extracted using the ArcFace R18 backbone. The t-SNE plots showcase distinct patterns of embedding behavior across corruption types. Severe corruptions such as spatter and contrast cause substantial feature divergence and class collapse, consistent with the significant performance degradation observed for these corruptions. In contrast, corruptions like blur and JPEG compression result in minimal feature drift with considerable overlap between clean and corrupted embeddings. Noise-based corruptions primarily induce feature drift without complete class collapse, maintaining some clustering structure. These visual insights effectively complement our quantitative mCEI metric analysis, providing additional evidence for the diagnostic value of DecordFace in understanding model robustness across different corruption types.

### 5.4 Quantifying mVCE on Real Samples

To showcase how similar performance degradations occur when an image corruption occurs in the wild, we perform experiments on two datasets- CelebA (Liu et al., 2015) and DecordFace dataset (Manchanda et al., 2023).

**The CelebA dataset (Liu et al., 2015).** An evaluation set was created using images from the CelebA dataset. One of the attributes annotated in the CelebA face images is blurry. We filtered all images from the dataset which were blurry. Then, using the identities of these subjects, we sampled face images that were 'not blurry.' Using these two sets, we created a clean subset of images with 6000 pairs (3000 genuine, 3000 impostors) where no face images were blurry. This is the CelebA Clean set. Similarly, using the same identities, we created a blurry variant where one of the images is blurry. This is the CelebA Blurry set containing 6000 verification pairs (3000 genuine, 3000 impostor). This experiment is conducted to showcase how the observations made using synthetic noises transfer well when the noise is environmental. Table 4 showcases an increase in mVCE when the images are blurry. This signifies the impact of image degradations on model performance.

**The D-LORD dataset (Manchanda et al., 2023).** To strengthen our evaluation, we additionally experiment with a subset of the D-LORD dataset, which includes faces captured

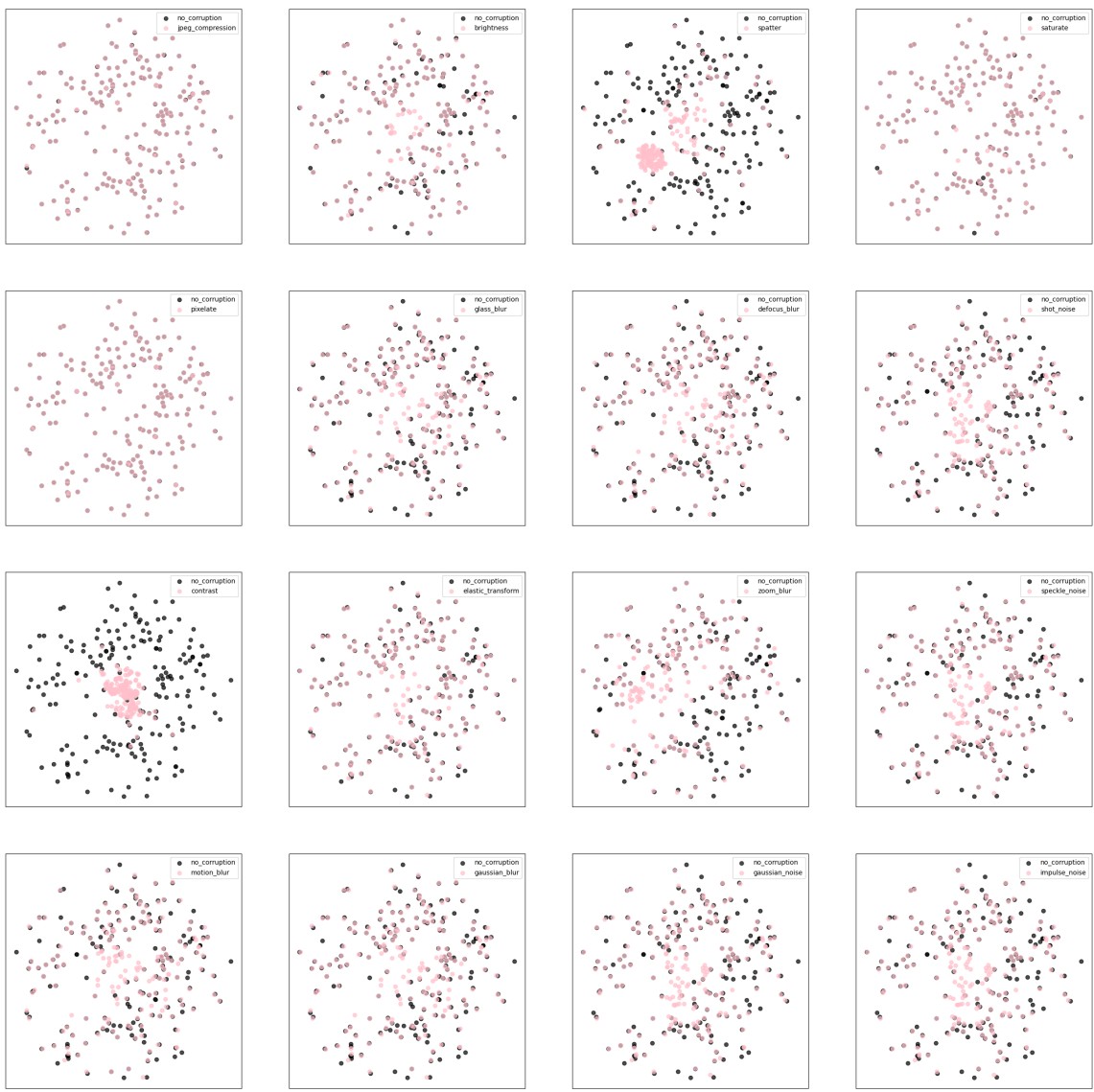

Figure 12: The t-SNE visualizations generated using 200 random samples from the AgeDB dataset corrupted at severity level 5, with features extracted using the ArcFace R18 backbone.

at increasing distances (5m, 7m, 10m, 15m) from the camera. Distance introduces natural degradation due to resolution loss and atmospheric effects, making it a relevant real-world proxy. We evaluated model performance using 6,000 pairs (3,000 genuine, 3,000 impostor) across the four distance settings. Error rates, computed as 1-TPR@FPR (FPR=0.0001), demonstrate consistent performance degradation with increasing distance across all evaluated models. The results (Table 5) confirm that real-world degradations produce significant robustness failures similar to those observed with synthetic corruptions. The findings reveal that smaller backbone architectures (LightCNN, ArcFace R18) experience sharp performance drops, while larger models (CosFace R100, AdaFace iR100) maintain greater resilience but still exhibit increasing error rates with distance. Importantly, these degradation patterns closely mirror the trends observed in our synthetic corruption experiments, supporting the predictive validity of DecordFace for real-world deployment scenarios. This evaluation demonstrates that DecordFace's diagnostic framework effectively translates to authentic degradation contexts, validating our approach despite the inherent challenges in acquiring comprehensive real-world corruption datasets. The consistency between synthetic and real-world results reinforces the practical utility of our benchmark for understanding model robustness characteristics.

## 5.5 Analyzing the Impact of Model Architecture and Training Method on Performance

In this section, we discuss the relationship between different FR algorithms and their performance and how factors such as their design, backbone size, and training data may play a role. The most obvious trend based on performance evaluations presented in Table 2 is that bigger models are more robust in the face of corruptions when compared to their smaller counterparts. This is true across all corruption severities. For example, in the ArcFace model, the mVCE increases with a decrease in backbone size for all datasets. The same can be seen to hold true for the CosFace and AdaFace models. The presence of more parameters enables the learning of more features and, therefore, leads to a more powerful and robust face feature embedding. Scaling has been shown as an effective way to improve model robustness. At the same time, it highlights the need for creating more robust, smaller models, which are essential for deployment on edge devices. In agreement with the mVCE values, the mCEI metric also shows better feature embedding invariance in the case of larger backbones.

Among all the models, the iR100 model backbone using the AdaFace algorithm provides the lowest mVCE values (Table 9). This is consistent with the performance comparison depicted by the authors in their paper for the quality adaptive recognition algorithm. Comparing the performance on AgeDB-*decord* dataset from Table 2, AdaFace iR100 trained on MSIMV3 provides an mVCE of 8.60% while the ArcFace R100 trained on the same dataset provides an mVCE of 9.53%. The same observations can be made between the Elastic-Face models and the iR100 backbone for AdaFace pretrained on MSIMV2 for all evaluation datasets. Further, it is interesting to observe that the AdaFace models trained on Web-Face4M/12M datasets provide better performance on the framework when compared to those trained on MSIMV2/V3 despite being trained on the same iR100 backbone. Web-Face4M/12M is the largest among the datasets used for pre-training in the selected models.

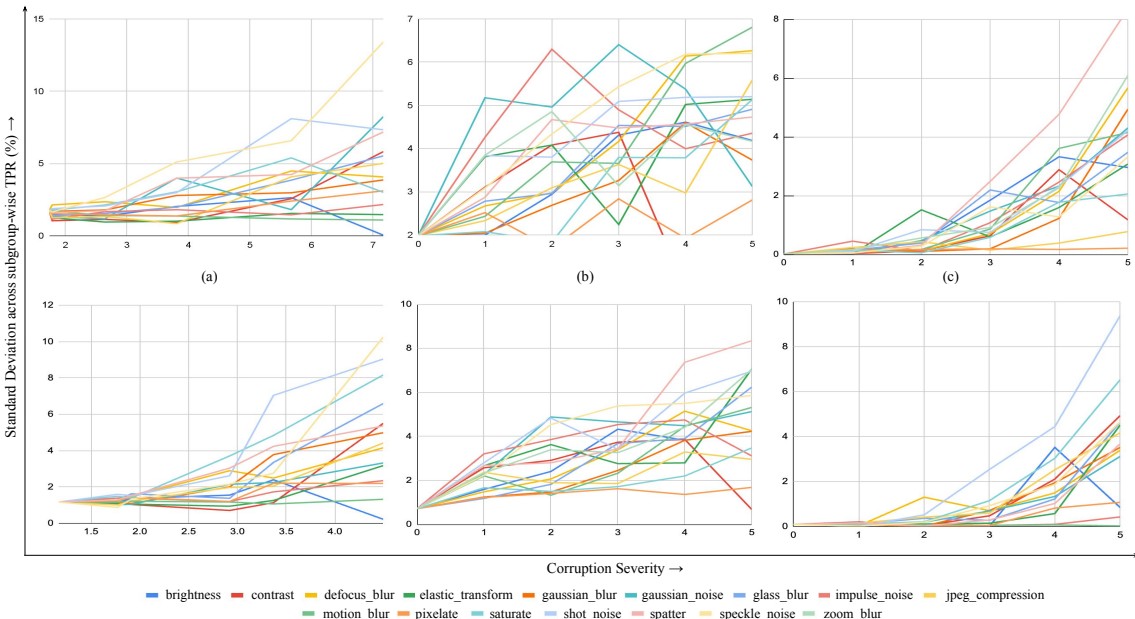

Figure 13: Plots highlighting the relationship between the evaluated models and their % drop in verification performance due to different corruptions on male and female subgroups in the (a) AgeDB-*decord* (b) CPLFW-*decord* and (c) CALFW-*decord* datasets. The standard deviation between the verification performance of subgroups on ArcFace ResNet50 backbone (top row) and ArcFace ResNet100 backbone (bottom row) are reported for different DecordFace subsets over all severities. The increasing standard deviation showcases the gap between performance on male and female subgroups increases with increasing corruption severity.

This aligns with the idea that larger training datasets lead to better models as they see more variation in data during training. Overall, we observe that the models trained on CASIA-WebFace perform poorly based on the performance comparison between iR50 backbones for AdaFace. This is also observed in the performance of the LightCNN model.

## 5.6 Analyzing the Performance of Models across Demographic Subgroups

In this section, we perform an analysis to understand the influence of corruption across different demographic subgroups. We divide the verification pairs for each dataset based on gender using the annotations provided in their respective datasets. Next, we evaluate these pairs separately using the different models. The TPR values for each subgroup are computed as per the evaluation protocol for the corresponding dataset. From Fig. 11(a) and (b), we observe that male and female subgroups are impacted differently by certain corruptions. The female subgroup is impacted more by nearly all corruptions, with an

Table 4: mVCE (%) on the clean and blurry subsets from the CelebA dataset showcasing a drop in model performance for these subsets.

| Model Name | Backbone | Clean | Blurry |
|---|---|---|---|
| LightCNN | 9L | 79.80 | 85.17 |
| LightCNN | 29L | 72.54 | 83.17 |
| LightCNN | 29Lv2 | 71.90 | 79.70 |
| CosFace | R18 | 63.04 | 66.04 |
| CosFace | R34 | 37.87 | 41.64 |
| CosFace | R50 | 22.44 | 27.84 |
| CosFace | R100 | 15.90 | 18.37 |
| ArcFace | R18 | 70.77 | 73.67 |
| ArcFace | R34 | 62.10 | 69.90 |
| ArcFace | R50 | 61.30 | 69.54 |
| ArcFace | R100 | 42.47 | 48.14 |
| MagFace | iR100 | 21.34 | 26.90 |
| ElasticFace-Arc | iR100 | 19.24 | 25.87 |
| ElasticFace-Cos | iR100 | 20.54 | 27.74 |
| ElasticFace-Arc+ | iR100 | 20.10 | 25.64 |
| ElasticFace-Cos+ | iR100 | 19.37 | 23.67 |
| AdaFace | iR18 (VGGFace2) | 59.90 | 61.77 |
| AdaFace | iR18 (CASIA) | 75.04 | 79.84 |
| AdaFace | iR50 (CASIA) | 62.94 | 74.94 |
| AdaFace | iR50 (MSIMV2) | 32.20 | 41.20 |
| AdaFace | iR100 (MS1MV2) | 17.10 | 21.54 |
| AdaFace | iR100 (MS1MV3) | 32.94 | 36.90 |
| AdaFace | iR18 (WebFace4M) | 59.20 | 64.27 |
| AdaFace | iR50 (WebFace4M) | 25.90 | 29.64 |
| AdaFace | iR100 (WebFace4M) | 12.57 | 15.44 |
| AdaFace | iR100 (WebFace12M) | 10.07 | 12.60 |
| CFSM-Arc | iR100 | 34.97 | 41.27 |

Table 5: Results showcasing the performance of different models on the D-LORD dataset, highlighting how model performance worsens with increasing degradations in real-world settings.

| Model Name | Backbone | 5m | 7m | 10m | 15m |
|---|---|---|---|---|---|
| LightCNN | 9L | 40.47 | 58.57 | 62.00 | 59.63 |
| LightCNN | 29L | 24.10 | 49.07 | 54.80 | 51.73 |
| LightCNN | 29Lv2 | 19.40 | 39.37 | 45.97 | 57.57 |
| CosFace | R18 | 38.90 | 55.67 | 62.97 | 64.23 |
| CosFace | R34 | 23.40 | 51.97 | 49.17 | 60.20 |
| CosFace | R50 | 17.87 | 26.13 | 43.60 | 48.40 |
| CosFace | R100 | 9.73 | 17.40 | 31.20 | 38.90 |
| ArcFace | R18 | 58.73 | 68.10 | 65.60 | 86.77 |
| ArcFace | R34 | 53.60 | 67.67 | 65.97 | 68.37 |
| ArcFace | R50 | 47.07 | 54.20 | 63.63 | 65.13 |
| ArcFace | R100 | 34.60 | 38.23 | 54.73 | 59.73 |
| MagFace | iR100 | 19.63 | 32.80 | 42.87 | 52.37 |
| ElasticFace-Arc | iR100 | 21.43 | 31.20 | 51.00 | 49.70 |
| ElasticFace-Cos | iR100 | 19.37 | 32.30 | 46.07 | 46.93 |
| ElasticFace-Arc+ | iR100 | 19.77 | 34.17 | 46.30 | 46.30 |
| ElasticFace-Cos+ | iR100 | 16.53 | 29.47 | 45.60 | 45.70 |
| AdaFace | iR18 (VGGFace2) | 50.33 | 48.57 | 66.83 | 70.97 |
| AdaFace | iR18 (CASIA) | 78.40 | 88.97 | 87.03 | 88.83 |
| AdaFace | iR50 (CASIA) | 57.77 | 74.73 | 74.37 | 72.10 |
| AdaFace | iR100 (MS1MV2) | 19.60 | 25.87 | 48.33 | 50.03 |
| AdaFace | iR100 (MS1MV3) | 27.70 | 36.63 | 44.57 | 53.93 |
| AdaFace | iR18 (WebFace4M) | 33.33 | 56.30 | 57.30 | 58.87 |
| AdaFace | iR50 (WebFace4M) | 27.80 | 29.00 | 44.10 | 57.30 |
| AdaFace | iR100 (WebFace4M) | 13.00 | 19.77 | 36.20 | 36.60 |
| AdaFace | iR100 (WebFace12M) | 11.10 | 16.63 | 36.13 | 35.80 |
| CFSM-Arc | iR100 | 28.07 | 34.40 | 54.23 | 51.40 |

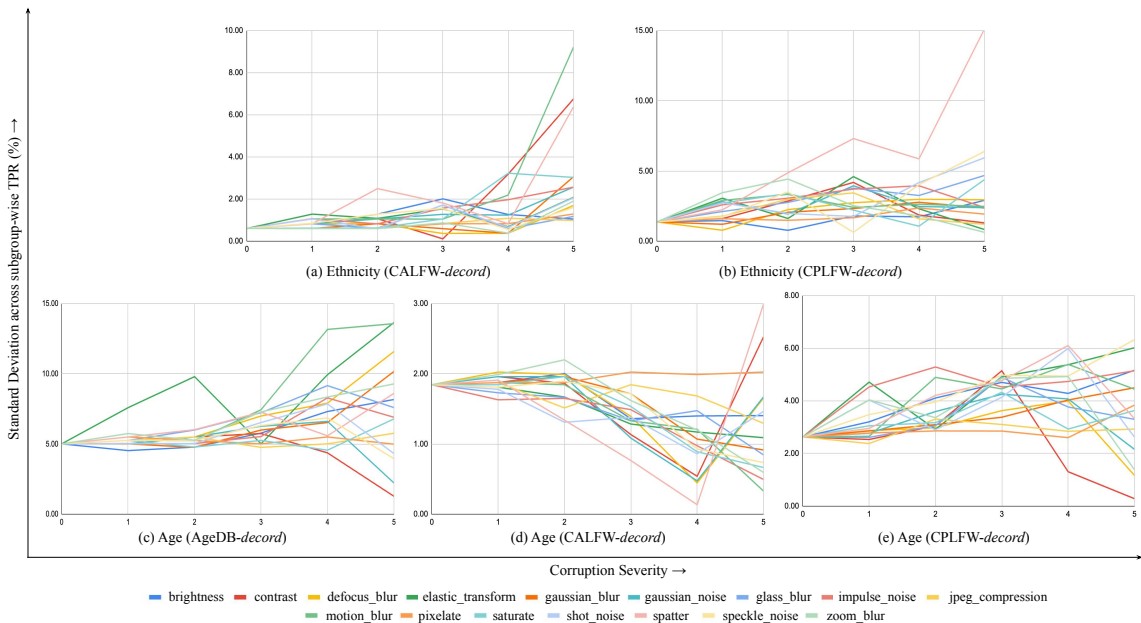

Figure 14: Plots highlighting the relationship between the evaluated models and their % drop in verification performance due to different corruptions on (a-c) young and old (age) subgroups, and (d-e) White, Black, and Asian (ethnicity) subgroups. The standard deviation between the verification performance of subgroups on ArcFace ResNet50 backbone is reported for different DecordFace subsets over all severities. The increasing standard deviation indicates a widening performance gap across ethnicity subgroups with increasing corruption severity, while the gap across age subgroups remains largely consistent.

overall performance degradation greater than 15% for more than 25% of models. On the other hand, the verification performance of the male subgroup is much less severely affected. Notably, for spatter, the female subgroup is impacted far more than the male subgroup, indicating occlusion to be an important factor in the misclassification of females. Similar observations can be made for blur-based and noise-based corruptions, indicating less robust learning of facial features associated with females. This could arise due to low variability in the dataset on which the algorithms are trained.

Since the model performance is also impacted by the severity of the corruptions, we compute the TPR values of the different subgroups and compute the standard deviation between the TPR values obtained for males and females. Standard deviation is a standard mechanism used in fairness studies to estimate the difference in verification performance across different subgroups. In Fig. 13, we plot this standard deviation between the verification performance (%) of male and female subgroups, and show trends on the AgeDB-*decord*, CPLFW-*decord*, and CALFW-*decord* datasets. An interesting pattern is observed where the standard deviation between the performance of the subgroups increases with an increase

in corruption severity, which is in agreement with existing work (Majumdar et al., 2021). This is an alarming observation since a high-performing model may break down for certain gender subgroups in the presence of unseen common image corruptions. The gap in performance between the subgroups can also be explored through the mVCE and mCEI metrics by calculating them separately for the male and female subgroups.

Similarly, we perform analysis for age and ethnicity subgroups. For these, we utilize the Fairface classifier(Karkkainen and Joo, 2021) to obtain age and ethnicity classes for the test pairs (except for AgeDB, for which ground truth age annotations are readily available). The age subgroups are divided into two classes- pairs between face images of people below the age of 50 (young) and above the age of 50 (old). The experiments are performed for the AgeDB, CALFW, and CPLFW variants of DecordFace. The results are shown in Figure 14(c-e). For the different age groups, we observe that the performance of the model varies significantly without any corruptions (∼10% for AgeDB), and the performance variation with corruptions does not follow a clear trend. For blur-based variations, the gap increases significantly between old and young subgroups for AgeDB30, with the model performing better for older subgroups. This is because AgeDB consists of test pairs with a 30-year gap. Therefore, the young pairs consist of samples matching younger faces to their adult versions, leading to lower performance. This gap is further exacerbated by blur-based corruption. In CPLFW, a similar trend is observed, however, the performance gap between subgroups is not as significant. In CALFW, the model performs better for younger test pairs as compared to old test pairs, signifying no clear trend in model performance. The performance gap is also less significant in this case.

For ethnicity, we considered the subgroups of Black, White, and Asian for the CALFW and CPLFW datasets. These three ethnicities were selected since there were 100+ genuine and impostor pairs for each of them ethnicities in the datasets. The results for the standard deviation are shown in Figure 14(a-b). Without any corruptions, the model performance across the three subgroups is observed to be consistent for CALFW, while for pairs in CPLFW, the performance on White pairs is slightly higher than that on Asian and Black pairs. The performance on Black pairs degrades the least for the spatter corruption. This is observed for both CALFW, and CPLFW pairs, showcasing the robustness of the model towards occlusion for faces belonging to the Black ethnicity. There are discrepancies in model performance degradation across corruption severities. However, these differences fall within the standard deviation of <3% for CALFW-decord and <5% for CPLFW-decord.Further exploration of fairness under common image corruptions can be considered as future work.

### 5.7 Analyzing the Performance of FROM (Qiu et al., 2021b)

Towards the robustness of FR models, there have been some approaches focusing on tackling the impact of occlusion on model performance (Song et al., 2019; Qiu et al., 2021a). These algorithms utilize trainable masks to discard non-meaningful features from the images. We utilize FROM (Qiu et al., 2021a) to showcase performance under corruptions on the AgeDB and CFP-FP datasets. The authors provide the pretrained models for the baseline and proposed method[9].

---

9. `https://github.com/haibo-qiu/FROM/`

Table 6: The mVCE performance (%) on the AgeDB-decord and CFP-FP-decord sets from the DecordFace framework on the different corruption severity protocols for the FROM algorithm (Qiu et al., 2021a). FROM uses the iR50 model backbone and the CASIA dataset for training.

| Dataset | mVCE (low) | mVCE (high) | mVCE (overall) |
|---------|-----------|-------------|----------------|
| AgeDB | 44.17 | 66.32 | 53.03 |
| CFP-FP | 18.53 | 39.67 | 26.98 |

In Table 6, the performance of FROM is provided for the low severity and high severity protocols, as well as the mVCE on all the corruption severities. The results indicate that the FROM algorithm also suffers from performance degradation in the presence of corruptions. Specifically, the model suffers severely for the AgeDB dataset, consisting of age variations (comparing performance with Table 9). The clean mVCE for the AgeDB dataset is 32.10% while the overall mVCE obtained from FROM is 53.03%. On the other hand, the model performance on CFP-FP varies from 13.06% on clean to 26.98% on the corrupted set. The FROM algorithm is trained on the CASIA dataset and using the iR50 model backbone. For the CFP-FP dataset, the performance degradation is comparable to other models, while for the AgeDB, the performance drop is significant, highlighting the algorithm is more robust to pose variations compared to age variations.

We also evaluate the baseline model utilized in the evaluation of FROM algorithm. While this baseline model also suffers from performance degradation under corruption, interesting trends appear. The FROM model appears to degrade less on the *spatter* corruption for the high severity protocol. Notably, the $VCE_{spatter}$ for the baseline is 77.87% while FROM is 71.27%. This is expected as the FROM algorithm primarily deals with occlusion and *spatter* is an occlusion-based corruption where random parts of the input image are removed. On the other hand, FROM performs significantly worse on all noise-based corruptions, losing robustness for other types of corruptions.

With this experiment, we observe that models trained for occlusion lead to performance degradation on other types of corruptions. While there is lesser performance degradation on occlusion-based corruptions like *spatter*, other corruptions still lead to large performance errors.

### 5.8 Denoising Baseline Evaluation using Restormer (Zamir et al., 2022)

We include a defense mechanism to demonstrate the practical utility of DecordFace for improving robustness. Specifically, we conducted experiments using Restormer (Zamir et al., 2022), a state-of-the-art transformer-based denoiser known for handling diverse corruptions efficiently, making it suitable for test-time integration. We employed its 'Real Denoising' variant to handle unknown distortions without retraining. The Restormer model is applied to images before face verification on the AgeDB30 dataset, following the same evaluation protocol as Table 2. The mVCE results (Table 7) show-

- Smaller backbones (e.g., LightCNN 9L, CosFace R18): denoising reduced error by up to 4%, indicating improved robustness.

- Intermediate models (e.g., R34): gains were minimal or inconsistent.

Table 7: Results showcasing the performance of models after images are denoised using Restormer(Zamir et al., 2022) before evaluation using the DecordFace framework.

| Model Name | Backbone | AgeDB | AgeDB + RestFormer |
|---|---|---|---|
| LightCNN | 9L | 60.40 | 56.73 |
| LightCNN | 29L | 43.54 | 43.76 |
| LightCNN | 29Lv2 | 37.55 | 38.11 |
| CosFace | R18 | 27.74 | 25.29 |
| CosFace | R34 | 14.24 | 15.11 |
| CosFace | R50 | 10.60 | 12.32 |
| CosFace | R100 | 9.28 | 10.99 |
| ArcFace | R18 | 27.61 | 27.10 |
| ArcFace | R34 | 16.62 | 16.68 |
| ArcFace | R50 | 11.80 | 13.27 |
| ArcFace | R100 | 9.53 | 10.78 |
| MagFace | iR100 | 13.60 | 13.52 |
| ElasticFace-Arc | iR100 | 13.63 | 12.61 |
| ElasticFace-Cos | iR100 | 14.69 | 14.15 |
| ElasticFace-Arc+ | iR100 | 14.02 | 13.39 |
| ElasticFace-Cos+ | iR100 | 15.84 | 14.62 |
| AdaFace | iR18 (VGGFace2) | 48.01 | 46.01 |
| AdaFace | iR18 (CASIA) | 54.34 | 52.24 |
| AdaFace | iR50 (CASIA) | 40.76 | 41.28 |
| AdaFace | iR100 (MS1MV2) | 10.08 | 9.99 |
| AdaFace | iR100 (MS1MV3) | 8.60 | 9.34 |
| AdaFace | iR18 (WebFace4M) | 31.35 | 31.80 |
| AdaFace | iR50 (WebFace4M) | 12.89 | 14.23 |
| AdaFace | iR100 (WebFace4M) | 9.19 | 10.95 |
| AdaFace | iR100 (WebFace12M) | 7.59 | 8.98 |
| CFSM-Arc | iR100 | 26.65 | 15.21 |

- Larger backbones (e.g., R50, R100): denoising slightly degraded performance, likely because these models already encode robust identity patterns, and Restormer-induced distribution shifts move samples away from known representations.

- An exception was CFSM-Arc, which exhibited substantial improvement (26.65% → 15.21% mVCE), highlighting model-specific responses to preprocessing interventions.

These findings suggest that denoising is beneficial for models with limited generalization capacity but can be counterproductive for more powerful models that rely on fine-grained identity features. Our hypothesis is that smaller models benefit when denoising pushes samples toward learned patterns, while larger models may be adversely affected by representation drift.

## 6 Discussion and Future Work

This work analyzes the limitations of current FR models in handling image degradations and corruptions. While existing work in the domain of Face Recognition has not emphasized building algorithms robust towards degradations, various works focus on image corruptions in the Computer Vision literature. However, these works focus largely on a classification setting with a limited number of classes, while in FR, the number of classes is generally large, and the models are expected to work in an open recognition setting.

In this section, we describe potential methods for solving this problem based on existing work and our understanding and knowledge of face recognition models.

**Adversarial Training:** Adversarial training in machine learning and deep learning refers to training a model on 'adversarial' examples for robust design. By learning adversarial samples, the model circumvents a distribution shift at test time. In the DecordFace framework, this would entail training the model on 'corrupted' samples of the data. This

is the simplest solution to this problem. However, this raises the question of other 'unseen' corruptions which are likely to occur in deployment settings.

**Image Restoration with Corruption Detection:** Another solution would involve utilizing a corruption detection module followed by an image enhancement method such as face restoration and super-resolution. Work corresponding to detection in faces has been conducted under face-spoofing detection, deepfake detection, attack detection, and face forgery detection (Chhabra et al., 2023; Thakral et al., 2023; Narayan and Patel, 2024). These methods may be extended to corruption. Some research has also been conducted into studying super-resolution and face restoration in conjunction with face recognition (Wang et al., 2022; Dosi et al., 2024). The performance of corruption detection in conjunction with image restoration will be dictated by the accuracy of the detection and restoration methods.

**Using AntiAliased Models during Training:** Zhang (2019) proposed the use of *blurpool* operation during the training of deep models. The rationale is that in signal processing, the fix for image corruptions is anti-aliasing by low-pass filtering before downsampling. Adapting FR models to incorporate the *blurpool* operation instead of the traditional pooling may enhance model robustness to corruptions. While this method would not require corrupted data for training, it would require re-training the FR models from scratch.

**LCANets in an FR Setting:** Teti et al. (2022) proposed LCANets which utilizes a spatiotemporal dictionary at the front of the model backbone to obtain robust features. These models are tested in an image recognition setting. Given the rich dictionary learning literature in the domain of face recognition (Manjani et al., 2017; Yadav et al., 2017), LCANets can be adapted to mitigate the impact of corruption in the FR setting. This method would also require training the FR models from scratch.

**Leveraging CEI for obtaining Compatible Embeddings:** In this work, we introduce the mCEI metric, which showcases the dissimilarity between the embedding of a clean image and its corrupted counterpart across the different models. For a given model, this metric stays consistent across the different datasets, highlighting stability in model behavior. By fine-tuning the model such that the invariance between the clean and corrupted embedding is optimized towards zero, it is possible to create a robust variant of the existing model. Existing work has shown that it is possible to obtain compatible embeddings across different models (Meng et al., 2021a), while no such research has been done for the same model but different distributions of data. Leveraging existing work along with our insights may aid in the development of a lightweight, robust FR algorithm.

In the future, exploring fairness mitigation strategies and adversarial robustness are both critical directions for responsible face recognition research. In this work, our primary goal is to diagnose and quantify robustness gaps under common real-world corruptions (e.g., blur, brightness, noise) using a standardized evaluation framework. Our framework is intended to serve as a diagnostic tool that can also be used to evaluate the effectiveness of future mitigation methods, including those addressing bias or adversarial threats.

While our contributions in this work remain diagnostic, the framework and corruptions in DecordFace could be adapted into training pipelines for face recognition models in future work. DecordFace's diagnostic results can directly inform targeted augmentation strategies, reinforcing its broader value, as these augmentations could also be combined to further enhance model robustness. However, it should be noted that if DecordFace is used for

training, models should be evaluated on datasets outside the framework to ensure fair assessment and avoid bias from reusing the same corruptions seen during training.

## 7 Summary

In this work, we introduce the DecordFace framework. The framework utilizes several existing FR datasets and creates their corrupted versions. We propose two important evaluation metrics, namely the mean Verification Corruption Error (mVCE) and mean Corruption Embedding Invariance (mCEI) metric, for the quantitative evaluation of performance degradation in the presence of corruptions. Based on thorough experimentation using over 25 pre-trained models across different architectures, algorithms, and pre-trained datasets, we summarize our observations as follows,

- The current deep learning models strongly suffer from performance degradation in the presence of common corruptions, especially when the corruptions are of high severity. In the DecordFace framework, we observed high-performance degradation under the *high-severity corruption protocol* across the board.

- Smaller models are far more prone to failure, even in the presence of less severe corruptions. While larger models performed well in the presence of *low severity* corruptions, lighter models, such as those with a ResNet18 or iResNet18 backbone, suffered greatly even in the presence of low severity corruptions. This introduces the important problem of developing smaller, more robust FR models.

- In the fairness evaluation, we observe a disparate performance across gender subgroups with an increase in corruption severity. This emphasizes the need for fairer *and* more robust models.

We believe that our framework opens important and exciting new challenges in face recognition, and would aid in the building of robust FR models in the future. Future research may include the exploration of performance degradation compounded by combining the different corruption types.

**Computational Requirements:** We acknowledge that a comprehensive robustness evaluation across multiple models and corruption settings can indeed be resource-intensive. To address this concern and ensure broader accessibility, we have designed DecordFace with modularity as a core principle. The framework allows researchers to conduct targeted evaluations based on their computational constraints through several mechanisms:

- Selective evaluation options: Users can choose specific subsets of models or corruptions most relevant to their research objectives, rather than requiring full-scale evaluation. For instance, researchers can demonstrate method efficacy using lightweight backbone models (e.g., LightCNN) without necessitating evaluation on larger architectures.

- Scalable corruption generation: Our tools support selective generation of corrupted datasets, enabling focused experiments on specific degradation types or severity levels rather than the complete corruption suite.

- Ready-to-use resources: We provide comprehensive evaluation scripts and dataset generation tools that significantly reduce setup overhead and enable efficient reproduction of results or targeted experiments.

These design choices ensure that DecordFace remains accessible to researchers with varying computational resources while preserving the diagnostic value of systematic robustness evaluation.

**Real World Applicability:** We clarify that experiments on DecordFace do not perfectly predict real-world performance. However, they support the view that robustness degradations exposed by DecordFace are indicative of vulnerabilities that also manifest under real-world shifts. The benchmark is valuable not because it reproduces real noise exactly, but because it offers a controlled diagnostic setting that generalizes well, especially when models have not been trained on the same noise patterns. We strongly encourage that the insights be integrated alongside comprehensive real-world evaluations, as much as possible.

## Broader Impact Statement

The DecordFace framework presents both positive and negative societal consequences stemming from its implementation in face recognition (FR) technologies. On the positive side, the framework enhances the reliability and robustness of FR models, which can significantly improve safety and security in critical applications such as surveillance and access control. Quantifying model performance under various image corruptions enables developers to create systems that maintain high accuracy in real-world conditions, thus enhancing user experience in applications like mobile device unlocking and social media tagging. Furthermore, the framework's focus on analyzing model performance across different demographic groups can contribute to reducing biases, leading to more equitable FR systems that serve all users fairly.

Conversely, the deployment of advanced FR technologies raises serious concerns regarding privacy and potential misuse (Mittal et al., 2024). While robust FR models can enhance security, they also increase the risk of surveillance and the erosion of personal privacy, particularly if misused by governments or other entities. We build DecordFace using existing face recognition datasets. All the face images in the paper are either taken or generated from face images present in existing datasets. As part of the benchmark, we release the scripts for generation.

Additionally, if the insights from the framework are not carefully considered, there is a risk of perpetuating existing biases, as systems may still perform inadequately for certain demographic groups. We acknowledge that performance disparities across gender and ethnicity in face recognition systems have serious societal implications, including risks of discrimination and unequal treatment. Therefore, there is an urgent need for fairness-aware design and evaluation in face recognition. Our framework is intended as a diagnostic tool to provide practitioners with a structured way to measure and expose such disparities. Further, a standardized bias mitigation protocol, built upon the insights provided by DecordFace, would be valuable for guiding industry practices in responsible deployment. To mitigate

these risks, it is also essential to establish ethical guidelines and regulations governing the use of FR technologies. We present some of these recommendations below.

- There is a need for measuring the robustness of FR models at deployment time to ensure sensitivity to different image corruptions.

- There is a need for measuring corruption-based robustness across different demographic subgroups, such as those of gender, ethnicity, and age.

- For building robust algorithms, the DecordFace framework may be employed for some datasets as a valuation step during training.

Overall, while the DecordFace framework has the potential to drive significant advancements in FR systems, it is crucial to navigate the associated ethical implications to ensure that these technologies benefit society as a whole. Our contributions are aimed at informing and enabling such future efforts, both in research and industrial settings.

## Acknowledgments and Disclosure of Funding

S. Mittal is partially supported by the IBM fellowship. M. Vatsa and R. Singh are supported through Srijan: COE on GenAI.

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

Table 8: The mVCE performance (%) on the DecordFace framework for the *low severity* corruption protocol.

| Model Name | Backbone | Pretraining Dataset | AgeDB -*decord* | CALFW -*decord* | CPLFW -*decord* | CFP-FP -*decord* | IJB-C-*decord* | | |
|---|---|---|---|---|---|---|---|---|---|
| | | | | | | | 1e-4 | 1e-5 | 1e-6 |
| LightCNN | 9L | MS-Celeb-1M + CASIA-WebFace | 32.46 | 34.04 | 67.77 | 44.85 | 28.23 | 40.08 | 52.79 |
| LightCNN | 29L | | 20.82 | 23.66 | 55.72 | 27.69 | 18.01 | 28.63 | 40.04 |
| LightCNN | 29Lv2 | | 16.48 | 20.31 | 50.01 | 23.61 | 15.84 | 29.94 | 54.15 |
| CosFace | R18 | Glint360k | 7.70 | 10.74 | 43.80 | 26.99 | 7.27 | 11.62 | 17.25 |
| CosFace | R34 | Glint360k | 4.46 | 8.33 | 29.69 | 16.69 | 4.80 | 7.66 | 13.40 |
| CosFace | R50 | Glint360k | 3.87 | 8.00 | 24.15 | 11.78 | 4.27 | 7.71 | 17.30 |
| CosFace | R100 | Glint360k | 3.46 | 7.72 | 20.69 | 7.87 | 4.24 | 8.70 | 23.68 |
| ArcFace | R18 | MS1MV3 | 7.87 | 11.22 | 47.49 | 34.72 | 8.16 | 12.17 | 17.80 |
| ArcFace | R34 | MS1MV3 | 4.84 | 8.61 | 36.67 | 25.37 | 5.94 | 9.10 | 15.17 |
| ArcFace | R50 | MS1MV3 | 3.97 | 8.05 | 31.64 | 20.39 | 5.10 | 7.78 | 12.92 |
| ArcFace | R100 | MS1MV3 | 3.48 | 7.76 | 26.48 | 17.59 | 4.43 | 7.00 | 13.15 |
| MagFace | iR100 | MS1MV3 | 3.86 | 8.03 | 26.83 | 15.14 | 5.76 | 9.02 | 14.65 |
| ElasticFace-Arc | iR100 | MS1MV2 | 3.81 | 7.97 | 26.76 | 14.72 | 5.40 | 8.37 | 15.78 |
| ElasticFace-Cos | iR100 | MS1MV2 | 4.20 | 7.85 | 26.69 | 14.34 | 5.02 | 8.04 | 14.38 |
| ElasticFace-Arc+ | iR100 | MS1MV2 | 3.78 | 7.92 | 26.90 | 14.38 | 5.30 | 8.10 | 13.49 |
| ElasticFace-Cos+ | iR100 | MS1MV2 | 4.26 | 7.96 | 26.24 | 15.12 | 5.10 | 8.21 | 15.43 |
| AdaFace | iR18 | VGGFace2 | 26.37 | 21.09 | 47.86 | 28.84 | 12.58 | 20.62 | 29.64 |
| AdaFace | iR18 | CASIA-WebFace | 31.27 | 33.33 | 97.16 | 44.84 | 89.76 | 99.63 | 99.98 |
| AdaFace | iR50 | CASIA-WebFace | 21.84 | 24.84 | 70.93 | 29.91 | 82.01 | 99.71 | 99.98 |
| AdaFace | iR50 | MSIMV2 | 4.91 | 8.22 | 29.77 | 16.50 | 6.38 | 12.04 | 20.84 |
| AdaFace | iR100 | MS1MV2 | 3.71 | 7.78 | 24.95 | 13.04 | 4.88 | 8.53 | 15.82 |
| AdaFace | iR100 | MS1MV3 | 3.79 | 7.64 | 25.34 | 16.10 | 4.09 | 6.77 | 12.00 |
| AdaFace | iR18 | WebFace4M | 14.98 | 13.74 | 42.75 | 27.92 | 8.11 | 13.30 | 19.77 |
| AdaFace | iR50 | WebFace4M | 5.62 | 8.83 | 25.07 | 13.70 | 4.06 | 6.63 | 11.42 |
| AdaFace | iR100 | WebFace4M | 4.47 | 8.16 | 18.97 | 7.67 | 3.33 | 5.46 | 9.22 |
| AdaFace | iR100 | WebFace12M | 3.81 | 7.80 | 17.75 | 7.57 | 2.88 | 4.95 | 13.22 |
| CFSM-Arc | iR100 | Cleaned MS1MV2 | 6.18 | 8.44 | 29.78 | 17.86 | 5.49 | 8.33 | 13.02 |

Table 9:  The mVCE performance (%) on the DecordFace framework across all severities.

| Model Name | Backbone | Pretraining Dataset | AgeDB -*decord* | CALFW -*decord* | CPLFW -*decord* | CFP-FP -*decord* | IJB-C-*decord* | | |
|---|---|---|---|---|---|---|---|---|---|
| | | | | | | | 1e-4 | 1e-5 | 1e-6 |
| LightCNN | 9L | MS-Celeb-1M + CASIA-WebFace | 43.63 | 43.26 | 73.79 | 52.96 | 41.21 | 52.70 | 64.36 |
| LightCNN | 29L | | 29.91 | 31.33 | 62.62 | 35.63 | 28.07 | 40.57 | 52.65 |
| LightCNN | 29Lv2 | | 24.91 | 27.93 | 57.95 | 31.43 | 25.73 | 43.24 | 65.55 |
| CosFace | R18 | Glint360k | 15.71 | 16.52 | 52.14 | 34.38 | 14.22 | 20.67 | 27.97 |
| CosFace | R34 | Glint360k | 8.37 | 11.52 | 36.59 | 22.14 | 8.52 | 13.63 | 21.42 |
| CosFace | R50 | Glint360k | 6.56 | 10.30 | 30.09 | 16.28 | 7.49 | 14.12 | 27.12 |
| CosFace | R100 | Glint360k | 5.79 | 9.77 | 26.28 | 12.24 | 7.62 | 15.29 | 34.56 |
| ArcFace | R18 | MS1MV3 | 15.77 | 17.22 | 54.38 | 41.58 | 13.98 | 19.93 | 27.13 |
| ArcFace | R34 | MS1MV3 | 9.55 | 12.38 | 42.70 | 30.82 | 10.18 | 15.14 | 23.39 |
| ArcFace | R50 | MS1MV3 | 7.10 | 10.66 | 36.95 | 25.27 | 8.41 | 12.47 | 18.89 |
| ArcFace | R100 | MS1MV3 | 5.90 | 10.01 | 31.46 | 21.66 | 7.43 | 11.36 | 19.35 |
| MagFace | iR100 | MS1MV3 | 7.75 | 11.13 | 33.58 | 20.89 | 11.51 | 16.95 | 24.44 |
| ElasticFace-Arc | iR100 | MS1MV2 | 7.74 | 11.58 | 33.47 | 20.83 | 10.68 | 15.81 | 25.18 |
| ElasticFace-Cos | iR100 | MS1MV2 | 8.39 | 11.24 | 33.11 | 20.02 | 9.77 | 15.12 | 24.22 |
| ElasticFace-Arc+ | iR100 | MS1MV2 | 7.88 | 11.26 | 33.01 | 20.00 | 9.92 | 14.52 | 22.27 |
| ElasticFace-Cos+ | iR100 | MS1MV2 | 8.90 | 11.58 | 32.81 | 20.87 | 10.08 | 15.39 | 25.02 |
| AdaFace | iR18 | VGGFace2 | 35.03 | 28.77 | 54.76 | 35.88 | 20.75 | 30.76 | 40.75 |
| AdaFace | iR18 | CASIA-WebFace | 40.50 | 40.82 | 97.75 | 52.90 | 92.97 | 99.74 | 99.99 |
| AdaFace | iR50 | CASIA-WebFace | 29.41 | 31.13 | 78.03 | 36.76 | 87.69 | 99.80 | 99.99 |
| AdaFace | iR50 | MSIMV2 | 8.35 | 10.57 | 35.71 | 21.50 | 12.56 | 20.48 | 30.87 |
| AdaFace | iR100 | MS1MV2 | 6.26 | 9.66 | 29.77 | 17.42 | 9.36 | 15.28 | 24.70 |
| AdaFace | iR100 | MS1MV3 | 5.72 | 8.95 | 29.47 | 19.42 | 6.07 | 9.99 | 16.63 |
| AdaFace | iR18 | WebFace4M | 21.48 | 18.54 | 49.02 | 33.30 | 13.03 | 20.45 | 28.89 |
| AdaFace | iR50 | WebFace4M | 8.53 | 10.86 | 29.05 | 16.73 | 5.44 | 9.16 | 15.62 |
| AdaFace | iR100 | WebFace4M | 6.36 | 9.35 | 22.17 | 10.18 | 4.22 | 7.05 | 12.01 |
| AdaFace | iR100 | WebFace12M | 5.32 | 8.72 | 20.70 | 9.91 | 3.70 | 6.60 | 16.71 |
| CFSM-Arc | iR100 | Cleaned MS1MV2 | 14.36 | 11.36 | 35.26 | 23.06 | 9.17 | 13.44 | 19.51 |

Table 10: The Relative mVCE performance (%) on the DecordFace framework for the *high severity* corruption protocol.

| Model Name | Backbone | Pretraining Dataset | AgeDB -*decord* | CALFW -*decord* | CPLFW -*decord* | CFP-FP -*decord* | IJB-C-*decord* | | |
|---|---|---|---|---|---|---|---|---|---|
| | | | | | | | 1e-4 | 1e-5 | 1e-6 |
| LightCNN | 9L | MS-Celeb-1M + CASIA-WebFace | 39.83 | 30.92 | 23.88* | 29.54 | 43.10 | 45.69 | 44.55 |
| LightCNN | 29L | | 29.54 | 25.34 | 26.2* | 27.00 | 31.31 | 39.26 | 43.41 |
| LightCNN | 29Lv2 | | 26.98 | 24.64 | 28.05* | 25.85 | 29.82 | 43.00 | 41.11 |
| CosFace | R18 | Glint360k | 23.04 | 16.46 | 28.95 | 22.72 | 19.68 | 26.68 | 32.21 |
| CosFace | R34 | Glint360k | 10.64 | 8.60 | 22.44 | 16.67 | 10.40 | 17.08 | 23.54 |
| CosFace | R50 | Glint360k | 7.43 | 6.27 | 18.53 | 13.92 | 9.05 | 18.69 | 31.31 |
| CosFace | R100 | Glint360k | 6.31 | 5.51 | 17.42 | 13.57 | 9.80 | 20.14 | 39.14 |
| ArcFace | R18 | MS1MV3 | 22.68 | 17.42 | 25.19 | 21.98 | 16.87 | 22.97 | 28.24 |
| ArcFace | R34 | MS1MV3 | 13.05 | 10.31 | 20.56 | 17.36 | 12.10 | 17.74 | 25.58 |
| ArcFace | R50 | MS1MV3 | 8.67 | 7.08 | 17.99 | 15.70 | 9.51 | 13.78 | 17.33 |
| ArcFace | R100 | MS1MV3 | 6.73 | 5.99 | 15.90 | 12.90 | 8.52 | 12.84 | 18.14 |
| MagFace | iR100 | MS1MV3 | 10.76 | 8.25 | 21.34 | 18.34 | 16.10 | 22.91 | 29.41 |
| ElasticFace-Arc | iR100 | MS1MV2 | 10.60 | 9.51 | 21.83 | 18.91 | 14.69 | 21.03 | 27.56 |
| ElasticFace-Cos | iR100 | MS1MV2 | 11.52 | 8.90 | 21.38 | 17.68 | 13.23 | 19.96 | 28.38 |
| ElasticFace-Arc+ | iR100 | MS1MV2 | 10.95 | 8.80 | 19.50 | 17.52 | 12.94 | 18.47 | 25.35 |
| ElasticFace-Cos+ | iR100 | MS1MV2 | 12.41 | 9.49 | 20.96 | 17.92 | 13.90 | 20.30 | 27.82 |
| AdaFace | iR18 | VGGFace2 | 28.41 | 23.15 | 23.75* | 22.64 | 24.35 | 31.43 | 35.40 |
| AdaFace | iR18 | CASIA-WebFace | 31.51 | 25.68 | 12.02* | 27.51 | 25.61 | 1.62* | 0.15* |
| AdaFace | iR50 | CASIA-WebFace | 25.43 | 19.73 | 27.04* | 22.03 | 36.11 | 1.22* | 0.11* |
| AdaFace | iR50 | MSIMV2 | 9.61 | 6.40 | 19.96 | 16.13 | 17.59 | 25.62 | 32.98 |
| AdaFace | iR100 | MS1MV2 | 7.11 | 5.08 | 15.80 | 13.76 | 12.64 | 19.48 | 26.06 |
| AdaFace | iR100 | MS1MV3 | 5.37 | 3.58 | 13.24 | 10.42 | 5.77 | 9.66 | 14.04 |
| AdaFace | iR18 | WebFace4M | 23.52 | 14.23 | 21.26 | 17.67 | 14.61 | 21.65 | 28.26 |
| AdaFace | iR50 | WebFace4M | 8.39 | 5.87 | 13.45 | 9.99 | 4.24 | 7.67 | 12.70 |
| AdaFace | iR100 | WebFace4M | 5.26 | 3.30 | 10.43 | 8.18 | 2.76 | 4.97 | 8.31 |
| AdaFace | iR100 | WebFace12M | 4.29 | 2.57 | 10.03 | 7.78 | 2.46 | 5.10 | 10.03 |
| CFSM-Arc | iR100 | Cleaned MS1MV2 | 23.02 | 7.91 | 17.50 | 16.16 | 10.42 | 14.82 | 19.51 |

Table 11: The Relative mVCE performance (%) on the DecordFace framework for the *low severity* corruption protocol.

| Model Name | Backbone | Pretraining Dataset | AgeDB -*decord* | CALFW -*decord* | CPLFW -*decord* | CFP-FP -*decord* | IJB-C-*decord* | | |
|---|---|---|---|---|---|---|---|---|---|
| | | | | | | | 1e-4 | 1e-5 | 1e-6 |
| LightCNN | 9L | MS-Celeb-1M + CASIA-WebFace | 11.89 | 7.87 | 8.84 | 9.25 | 10.66 | 14.14 | 15.63 |
| LightCNN | 29L | | 6.82 | 6.16 | 8.95 | 7.15 | 6.15 | 9.42 | 11.88 |
| LightCNN | 29Lv2 | | 5.91 | 5.58 | 8.21 | 6.30 | 5.10 | 9.77 | 12.60 |
| CosFace | R18 | Glint360k | 3.00 | 2.01 | 8.10 | 4.25 | 2.32 | 4.05 | 5.41 |
| CosFace | R34 | Glint360k | 0.86 | 0.63 | 5.19 | 3.03 | 1.09 | 2.16 | 3.48 |
| CosFace | R50 | Glint360k | 0.70 | 0.53 | 3.68 | 2.67 | 1.01 | 2.67 | 6.77 |
| CosFace | R100 | Glint360k | 0.49 | 0.39 | 3.46 | 2.64 | 1.35 | 3.68 | 11.95 |
| ArcFace | R18 | MS1MV3 | 2.94 | 2.42 | 7.96 | 4.83 | 2.31 | 3.57 | 4.92 |
| ArcFace | R34 | MS1MV3 | 1.27 | 0.88 | 5.50 | 3.74 | 1.48 | 2.63 | 5.02 |
| ArcFace | R50 | MS1MV3 | 0.84 | 0.55 | 4.71 | 3.50 | 1.22 | 2.05 | 2.40 |
| ArcFace | R100 | MS1MV3 | 0.68 | 0.36 | 3.45 | 2.73 | 1.01 | 1.94 | 2.64 |
| MagFace | iR100 | MS1MV3 | 1.03 | 0.50 | 4.46 | 3.97 | 1.73 | 3.10 | 4.93 |
| ElasticFace-Arc | iR100 | MS1MV2 | 0.78 | 0.50 | 5.06 | 3.63 | 1.48 | 2.44 | 4.06 |
| ElasticFace-Cos | iR100 | MS1MV2 | 1.03 | 0.42 | 5.32 | 3.48 | 1.34 | 2.27 | 3.78 |
| ElasticFace-Arc+ | iR100 | MS1MV2 | 0.71 | 0.45 | 4.23 | 3.47 | 1.38 | 2.43 | 3.41 |
| ElasticFace-Cos+ | iR100 | MS1MV2 | 0.83 | 0.43 | 4.54 | 3.55 | 1.43 | 2.34 | 3.83 |
| AdaFace | iR18 | VGGFace2 | 6.77 | 3.96 | 6.49 | 5.04 | 3.93 | 6.08 | 7.63 |
| AdaFace | iR18 | CASIA-WebFace | 8.44 | 6.96 | 10.56 | 7.35 | 17.59 | 1.36 | 0.14 |
| AdaFace | iR50 | CASIA-WebFace | 6.51 | 4.01 | 9.30 | 4.91 | 21.90 | 1.01 | 0.10 |
| AdaFace | iR50 | MSIMV2 | 1.01 | 0.52 | 5.10 | 3.64 | 2.15 | 4.53 | 7.91 |
| AdaFace | iR100 | MS1MV2 | 0.74 | 0.38 | 3.75 | 2.81 | 1.43 | 2.61 | 3.87 |
| AdaFace | iR100 | MS1MV3 | 0.56 | 0.31 | 2.91 | 2.10 | 0.82 | 1.59 | 2.45 |
| AdaFace | iR18 | WebFace4M | 7.15 | 2.24 | 5.58 | 4.21 | 2.31 | 3.78 | 5.47 |
| AdaFace | iR50 | WebFace4M | 1.12 | 0.80 | 3.50 | 2.41 | 0.80 | 1.34 | 2.18 |
| AdaFace | iR100 | WebFace4M | 0.54 | 0.34 | 2.44 | 1.90 | 0.53 | 1.00 | 1.32 |
| AdaFace | iR100 | WebFace12M | 0.51 | 0.27 | 2.65 | 1.94 | 0.42 | 0.98 | 1.30 |
| CFSM-Arc | iR100 | Cleaned MS1MV2 | 2.55 | 0.61 | 3.81 | 3.15 | 1.21 | 2.05 | 3.30 |

Table 12: The Relative mVCE performance (%) on the DecordFace framework over all corruption severities.

| Model Name | Backbone | Pretraining Dataset | AgeDB -decord | CALFW -decord | CPLFW -decord | CFP-FP -decord | IJB-C-decord | | |
|---|---|---|---|---|---|---|---|---|---|
| | | | | | | | 1e-4 | 1e-5 | 1e-6 |
| LightCNN | 9L | MS-Celeb-1M + CASIA-WebFace | 23.06 | 17.09 | 14.86 | 17.36 | 23.64 | 26.76 | 27.20 |
| LightCNN | 29L | | 15.91 | 13.83 | 15.85 | 15.09 | 16.21 | 21.36 | 24.49 |
| LightCNN | 29Lv2 | | 14.34 | 13.20 | 16.15 | 14.12 | 14.99 | 23.06 | 24.00 |
| CosFace | R18 | Glint360k | 11.01 | 7.79 | 16.44 | 11.64 | 9.27 | 13.10 | 16.13 |
| CosFace | R34 | Glint360k | 4.77 | 3.82 | 12.09 | 8.48 | 4.81 | 8.13 | 11.50 |
| CosFace | R50 | Glint360k | 3.39 | 2.83 | 9.62 | 7.17 | 4.23 | 9.08 | 16.59 |
| CosFace | R100 | Glint360k | 2.82 | 2.44 | 9.05 | 7.01 | 4.73 | 10.27 | 22.83 |
| ArcFace | R18 | MS1MV3 | 10.84 | 8.42 | 14.85 | 11.69 | 8.13 | 11.33 | 14.25 |
| ArcFace | R34 | MS1MV3 | 5.98 | 4.65 | 11.52 | 9.19 | 5.72 | 8.67 | 13.24 |
| ArcFace | R50 | MS1MV3 | 3.97 | 3.16 | 10.02 | 8.38 | 4.53 | 6.74 | 8.37 |
| ArcFace | R100 | MS1MV3 | 3.10 | 2.61 | 8.43 | 6.80 | 4.01 | 6.30 | 8.84 |
| MagFace | iR100 | MS1MV3 | 4.92 | 3.60 | 11.21 | 9.72 | 7.48 | 11.03 | 14.72 |
| ElasticFace-Arc | iR100 | MS1MV2 | 4.71 | 4.11 | 11.77 | 9.74 | 6.76 | 9.88 | 13.46 |
| ElasticFace-Cos | iR100 | MS1MV2 | 5.22 | 3.81 | 11.74 | 9.16 | 6.09 | 9.35 | 13.62 |
| ElasticFace-Arc+ | iR100 | MS1MV2 | 4.81 | 3.79 | 10.34 | 9.09 | 6.00 | 8.85 | 12.19 |
| ElasticFace-Cos+ | iR100 | MS1MV2 | 5.47 | 4.05 | 11.11 | 9.30 | 6.42 | 9.52 | 13.42 |
| AdaFace | iR18 | VGGFace2 | 15.43 | 11.64 | 13.39 | 12.08 | 12.10 | 16.22 | 18.74 |
| AdaFace | iR18 | CASIA-WebFace | 17.67 | 14.45 | 11.15 | 15.41 | 20.80 | 1.47 | 0.15 |
| AdaFace | iR50 | CASIA-WebFace | 14.08 | 10.30 | 16.40 | 11.76 | 27.58 | 1.10 | 0.11 |
| AdaFace | iR50 | MSIMV2 | 4.45 | 2.87 | 11.04 | 8.64 | 8.33 | 12.97 | 17.94 |
| AdaFace | iR100 | MS1MV2 | 3.29 | 2.26 | 8.57 | 7.19 | 5.91 | 9.36 | 12.75 |
| AdaFace | iR100 | MS1MV3 | 2.49 | 1.62 | 7.04 | 5.42 | 2.80 | 4.81 | 7.08 |
| AdaFace | iR18 | WebFace4M | 13.65 | 7.04 | 11.85 | 9.59 | 7.23 | 10.93 | 14.59 |
| AdaFace | iR50 | WebFace4M | 4.03 | 2.83 | 7.48 | 5.44 | 2.18 | 3.87 | 6.39 |
| AdaFace | iR100 | WebFace4M | 2.43 | 1.52 | 5.64 | 4.41 | 1.42 | 2.59 | 4.11 |
| AdaFace | iR100 | WebFace12M | 2.02 | 1.19 | 5.60 | 4.28 | 1.24 | 2.63 | 4.79 |
| CFSM-Arc | iR100 | Cleaned MS1MV2 | 10.73 | 3.53 | 9.29 | 8.35 | 4.89 | 7.16 | 9.79 |

Table 13: The mCEI performance (%) on the *low severity* corruption protocol.

| Model Name | Backbone | Pretraining Dataset | AgeDB -decord | CALFW -decord | CPLFW -decord | CFP-FP -decord | IJB-C -decord |
|---|---|---|---|---|---|---|---|
| LightCNN | 9L | MS-Celeb-1M + CASIA-WebFace | 86.15 | 85.99 | 84.23 | 84.17 | 86.59 |
| LightCNN | 29L | | 90.36 | 89.78 | 88.02 | 87.59 | 90.62 |
| LightCNN | 29Lv2 | | 91.07 | 90.63 | 88.54 | 89.01 | 91.65 |
| CosFace | R18 | Glint360k | 87.53 | 88.87 | 87.12 | 86.43 | 90.89 |
| CosFace | R34 | Glint360k | 89.11 | 90.63 | 89.21 | 88.74 | 92.40 |
| CosFace | R50 | Glint360k | 89.64 | 91.15 | 90.01 | 89.42 | 92.75 |
| CosFace | R100 | Glint360k | 89.64 | 91.80 | 90.77 | 90.09 | 92.07 |
| ArcFace | R18 | MS1MV3 | 88.81 | 88.29 | 85.96 | 85.50 | 90.42 |
| ArcFace | R34 | MS1MV3 | 90.06 | 89.77 | 87.33 | 87.09 | 91.74 |
| ArcFace | R50 | MS1MV3 | 90.57 | 90.27 | 88.03 | 87.51 | 92.28 |
| ArcFace | R100 | MS1MV3 | 90.97 | 90.68 | 88.20 | 87.92 | 92.72 |
| MagFace | iR100 | MS1MV3 | 91.53 | 91.40 | 88.93 | 89.28 | 92.92 |
| ElasticFace-Arc | iR100 | MS1MV2 | 90.65 | 90.51 | 87.40 | 87.90 | 91.99 |
| ElasticFace-Cos | iR100 | MS1MV2 | 89.67 | 89.34 | 86.91 | 87.02 | 91.22 |
| ElasticFace-Arc+ | iR100 | MS1MV2 | 90.47 | 90.28 | 87.33 | 87.73 | 91.92 |
| ElasticFace-Cos+ | iR100 | MS1MV2 | 89.27 | 88.90 | 86.83 | 86.71 | 91.05 |
| AdaFace | iR18 | VGGFace2 | 89.29 | 89.21 | 86.43 | 86.72 | 90.09 |
| AdaFace | iR18 | CASIA-WebFace | 87.88 | 87.14 | 84.65 | 85.31 | 88.66 |
| AdaFace | iR50 | CASIA-WebFace | 89.24 | 88.52 | 86.33 | 87.04 | None |
| AdaFace | iR50 | MSIMV2 | 91.05 | 90.49 | 88.31 | 88.39 | 91.72 |
| AdaFace | iR100 | MS1MV2 | 91.47 | 90.93 | 88.66 | 88.94 | 92.24 |
| AdaFace | iR100 | MS1MV3 | 91.39 | 90.96 | 89.20 | 88.66 | 92.79 |
| AdaFace | iR18 | WebFace4M | 90.22 | 89.56 | 88.47 | 87.64 | 91.06 |
| AdaFace | iR50 | WebFace4M | 91.92 | 91.48 | 90.79 | 89.70 | 93.61 |
| AdaFace | iR100 | WebFace4M | 92.51 | 92.03 | 91.36 | 90.37 | 94.28 |
| AdaFace | iR100 | WebFace12M | 92.30 | 91.74 | 91.07 | 90.01 | 94.04 |
| CFSM-Arc | iR100 | Cleaned MS1MV2 | 88.19 | 90.76 | 88.03 | 88.38 | 92.41 |

Table 14:   The mCEI performance (%) over all corruption severities.

| Model Name | Backbone | Pretraining Dataset | AgeDB -decord | CALFW -decord | CPLFW -decord | CFP-FP -decord | IJB-C -decord |
|---|---|---|---|---|---|---|---|
| LightCNN | 9L | MS-Celeb-1M + CASIA-WebFace | 74.39 | 74.31 | 72.94 | 72.77 | 75.91 |
| LightCNN | 29L | | 80.85 | 80.22 | 78.24 | 77.69 | 82.17 |
| LightCNN | 29Lv2 | | 81.81 | 81.37 | 78.80 | 79.42 | 83.42 |
| CosFace | R18 | Glint360k | 77.85 | 79.55 | 77.55 | 77.18 | 82.80 |
| CosFace | R34 | Glint360k | 81.13 | 82.82 | 80.56 | 80.45 | 85.92 |
| CosFace | R50 | Glint360k | 82.29 | 83.93 | 82.05 | 81.73 | 86.87 |
| CosFace | R100 | Glint360k | 82.55 | 84.83 | 83.09 | 82.66 | 86.48 |
| ArcFace | R18 | MS1MV3 | 79.09 | 78.63 | 76.04 | 75.63 | 81.96 |
| ArcFace | R34 | MS1MV3 | 81.28 | 81.29 | 77.93 | 78.05 | 84.18 |
| ArcFace | R50 | MS1MV3 | 82.46 | 82.43 | 79.29 | 78.98 | 85.37 |
| ArcFace | R100 | MS1MV3 | 83.25 | 83.14 | 79.66 | 79.75 | 86.20 |
| MagFace | iR100 | MS1MV3 | 83.06 | 83.06 | 79.55 | 80.31 | 85.38 |
| ElasticFace-Arc | iR100 | MS1MV2 | 81.41 | 81.40 | 77.17 | 78.17 | 83.62 |
| ElasticFace-Cos | iR100 | MS1MV2 | 80.09 | 79.80 | 76.62 | 77.05 | 82.68 |
| ElasticFace-Arc+ | iR100 | MS1MV2 | 81.35 | 81.28 | 77.40 | 78.19 | 83.85 |
| ElasticFace-Cos+ | iR100 | MS1MV2 | 79.50 | 79.16 | 76.42 | 76.65 | 82.47 |
| AdaFace | iR18 | VGGFace2 | 79.95 | 79.46 | 75.91 | 76.57 | 81.23 |
| AdaFace | iR18 | CASIA-WebFace | 77.53 | 76.31 | 73.43 | 74.66 | 79.02 |
| AdaFace | iR50 | CASIA-WebFace | 80.65 | 79.68 | 76.59 | 78.01 | 82.68 |
| AdaFace | iR50 | MSIMV2 | 82.76 | 82.15 | 79.00 | 79.38 | 84.09 |
| AdaFace | iR100 | MS1MV2 | 83.44 | 82.77 | 79.56 | 80.15 | 84.84 |
| AdaFace | iR100 | MS1MV3 | 84.17 | 83.76 | 81.11 | 80.80 | 86.63 |
| AdaFace | iR18 | WebFace4M | 81.25 | 81.05 | 79.77 | 78.91 | 83.53 |
| AdaFace | iR50 | WebFace4M | 85.36 | 84.82 | 83.80 | 82.51 | 88.39 |
| AdaFace | iR100 | WebFace4M | 86.39 | 85.84 | 84.82 | 83.69 | 89.62 |
| AdaFace | iR100 | WebFace12M | 86.06 | 85.36 | 84.28 | 83.09 | 89.17 |
| CFSM-Arc | iR100 | Cleaned MS1MV2 | 77.46 | 82.81 | 79.02 | 79.71 | 85.31 |

