# OpenReview forum: "DecordFace: A Framework for Degraded and Corrupted Face Recognition"
_DMLR — Accepted by DMLR_

### Review · Reviewer_n3mM · 2025-07-12

**Recommendation:** 1
**Confidence:** 3

**Summary Of Contributions:**

This paper proposes to create standardized corruptions of well known datasets for face recognition tasks in order to evaluate face recognition systems. Two new metrics are proposed to perform the evaluation. The corruptions are explicit and happen in 5 levels of severity for each corruption. These are: adding a blur (defocus, gaussian, glass, motion, zoom), adding noise (gaussian, impulse, shot, speckle), changing light and colour conditions (brightness, contrast, saturate), adding distortion (elastic transform), compression effects (JPEG compression, pixelate) and adding occlussions (spatter). The two metrics are specifically designed to be used as an evaluation set.
Code is provided that uses a combination of standard image transformation libraries (PIL, skyimage, wand.image, ...) and well defined mathematical terms in numpy for some operations.

**Strengths:**

* Well and clearly written.

**Audience:**

Yes

**Broader Impact Concerns:**

I have no concerns with the broader impact statement given.

**Claims And Evidence:**

As outlined above, I find the major premise, that performing an evaluation on a specific set of non-interacting image distortions of varying strengths on a given dataset to have a meaningful outcome as not proven. Sadly, this taints the entire paper.

**Datasets And Benchmarks:**

The dataset would consist of distortions generated by the code provided of already well known datasets. As such this process is highly reproducible. There are no concerns here.

**Extended Submissions:**

N/A

**Limitations:**

The major limitation is outlined already in the "weakness section". I cannot see how this is a reasonable evaluation framework that has any impact when performed. I can see value in it as a data augmentation process, but, at the moment, fail to see where this would otherwise help.

**Requested Changes:**

I would recommend major surgery on this paper. To me, this method should not be an evaluation framework but would have merit as a data augmentation process that could be applied to any dataset. The code base would require minimal changes (a standardized dataloader is nearly already present) and
it would make it more general. This would then leave a section to evaluate whether these pure, non-combined, distortions are having a proper real world effect in well known, real world face recognition evaluation benchmarks. I think some or most of this work is already present, but might need some work to recast the results along that line.
This leaves the question of what to do with the two derived new metrics. In their current form they would not fit.
Now, of course, there might be other ways to proceed with this paper and I would encourage the authors to pursue these. For me, at the moment though, there seems little value in this dataset as an evaluation framework for the reasons outlined in the "weakness section" above.

**Strengths And Weaknesses:**

Strengths:
* Comprehensive and well written paper outlining and documenting the idea.

Weaknesses:
* A glaring weakness for me, which I cannot ignore, lies in the basic idea itself. This paper enumerates particular, pure, distortions and uses these as the basis of an evaluation. These distortions are not combined, nor do they stand any chance of representing the subtle real world distortions a real world dataset will likely contain. In fact, working off real world data is precisely necessary, as real world distortions are mostly not following known distortion patterns, let alone pure and not combined ones as are presented here. To me, this paper misses the point: the derived dataset should be used as training data augmentation for certain applications which need to be robust against the relevant distortions. Whether this augmentation helps should then be evaluated on real world distortions. Put it in another way, if I was told that a model performed well wrt this evaluation, it would give me no confidence whatsoever that it would perform well in a real world setting. If a model were to perform poorly, it might be a good idea to retrain with data augmentation suggested here, but even then I would not be able to be clear whether it would perform poorly in a real world setting.
* The suggested alterations of the TPR@FPR metrics here are specifically done so this dataset could be used as an evaluation dataset. Given my last point, this renders them somewhat obsolete.

---

### Review · Reviewer_sqSN · 2025-07-17

**Recommendation:** 4
**Confidence:** 2

**Summary Of Contributions:**

The paper introduces DecordFace, a novel evaluation framework designed to systematically assess the robustness of face recognition (FR) systems under common image degradations and corruptions. The framework includes:

•	16 corruption types spanning noise, blur, compression, colour transformations, distortions, and occlusions, each evaluated at five severity levels.

•	Corrupted versions of five widely used face verification datasets, creating over 126 million verification pairs.

•	Two new evaluation metrics:

      o	Mean Verification Corruption Error (mVCE) quantifies performance degradation under corruption.

      o	Mean Corruption Embedding Invariance (mCEI) measures changes in feature space representations.

•	An extensive benchmarking study using 27 popular FR models with various backbones (ResNet, iResNet, LightCNN) under the DecordFace framework.

•	Key findings include a substantial performance drop (often over 20%) in the presence of noise, blur, and colour transformations, especially affecting shallow backbones and under certain demographic subgroups (e.g., gender).

The code and datasets are publicly available, promoting transparency and reproducibility.

**Strengths:**

•	Timely and Novel: First large-scale benchmark on real-world degradations in face recognition.

•	Comprehensive Evaluation: Covers 5 datasets, 16 corruption types, 5 severity levels, and 27 face recognition models.

•	New Metrics: Introduces mVCE and mCEI for both verification and feature-level analysis.

•	Open-Source: Dataset and code are publicly available for the community.

•	Demographic Insights: Shows how performance degrades differently across gender and demographic groups.

**Audience:**

Yes

**Broader Impact Concerns:**

Ethical implications of deploying FR systems with significant performance disparities across gender and ethnicity.

The necessity for a standardised bias mitigation protocol when using DecordFace in industry pipelines.
A stronger Broader Impact Statement could guide responsible use and mitigation planning for practitioners.

**Claims And Evidence:**

The claims are generally well-supported:

•	Empirical claims are validated on large-scale datasets with robust experimental protocols.

•	Metric definitions are mathematically sound and justified.

•	Claims around bias are statistically validated, albeit with some room for deeper subgroup analysis.

•	One area of caution is the assumption of transferability to real-world distortions, which is only partially validated.

**Datasets And Benchmarks:**

Sufficient Detail Present:

Clear description of data collection and the corruption process.

Open-source availability on GitHub.

**Extended Submissions:**

There is no evidence that this is an extended version of prior published work. The manuscript appears original, introducing new datasets, frameworks, and metrics.

**Limitations:**

•	Synthetic Bias: Heavy reliance on synthetic corruption may not fully reflect complex real-world image distortions.

•	Compute Intensity: The high resource demand makes replicability challenging for small-scale research environments.

•	Limited Downstream Tasks: The scope is restricted to verification, excluding identification and re-identification tasks prevalent in real-world use-cases.

•	No Adversarial Overlap: While focusing on common corruptions, the framework doesn’t integrate adversarial robustness perspectives.

**Requested Changes:**

Generalisation to Real-World Settings: Include additional natural degradation datasets or real-world deployment scenarios (beyond CelebA Blurry) to assess external validity.

mCEI Clarification: Clarify limitations of mCEI, especially where feature invariance does not guarantee classification accuracy.

**Strengths And Weaknesses:**

Strengths

•	Timely and Novel: First large-scale benchmark on real-world degradations in face recognition.

•	Comprehensive Evaluation: Covers 5 datasets, 16 corruption types, 5 severity levels, and 27 face recognition models.

•	New Metrics: Introduces mVCE and mCEI for both verification and feature-level analysis.

•	Open-Source: Dataset and code are publicly available for the community.

•	Demographic Insights: Shows how performance degrades differently across gender and demographic groups.


Weaknesses

•	Limited Real-World Testing: Mostly tested on simulated corruptions, with minimal validation on real-world corruptions.

•	Metric Assumptions: mCEI assumes feature stability always aligns with verification accuracy, which may not always be true.

•	High Computational Needs: Requires large compute resources, making it hard for smaller research teams to use.

•	Shallow Fairness Analysis: Identifies bias but does not explore ways to fix or mitigate it.

---

### Review · Reviewer_teh6 · 2025-07-17

**Recommendation:** 3
**Confidence:** 2

**Summary Of Contributions:**

This paper introduces DecordFace, a proposed benchmark framework for evaluating the robustness of face recognition models under common image degradations and corruptions. The authors generate corrupted variants of 5 standard FR datasets (i.e., AgeDB, CALFW, CFP-FP, CPLFW, and IJB-C) using 16 types of corruptions at 5 severity levels, which results in over 126 million pairs for evaluation. In addition, the authors evaluate 27 face recognition models with a diverse range of backbones and training setups using conventional verification metrics and propose two new metrics: mean Verification Corruption Error (mVCE) and mean Corruption Embedding Invariance (mCEI). The authors' analysis shows significant performance degradation under noise, blur, and contrast corruptions, especially for shallower models and certain demographic subgroups (e.g., females). The framework demonstrates certain fairness concerns, model architecture impacts, and the importance of robust training. Notably, all code and datasets are released publicly for reproducibility and open science purposes.

**Strengths:**

As discussed in previous box, this paper proposes a framework with relatively large in scale and has done a comprehensive evaluation over different models. This could benefit the researchers in this community and spur future research in this domain. Other strengths have been discussed in the previous box as well.

**Audience:**

Yes

**Claims And Evidence:**

Yes

**Datasets And Benchmarks:**

Yes

**Extended Submissions:**

N/A

**Requested Changes:**

1. The paper reveals robustness failures but does not explore or propose potential ways about how to address them. Consider including a baseline defense method such as test-time data augmentation, adversarial training, denoising autoencoders, or corruption-aware training to demonstrate the practical utility of DecordFace for improving robustness.
2. The authors explicitly exclude rain, snow, and frost due to "unrealistic appearance." However, these can occur in surveillance contexts and could have been optionally included, quantitatively justified, or at least discussed in more details.
3. To complement the mCEI metric, use dimensionality reduction (e.g., t-SNE or PCA) to visualize how embeddings of clean vs. corrupted images shift in latent space. This can help diagnose what types of corruptions cause class collapse or feature drift.
4. The paper avoids cross-validation-based threshold selection, opting for fixed thresholds. Provide additional quantitative or visual evidence (e.g., ROC curves for different thresholds) to show how this choice impacts fairness or generalization and whether it might bias against certain models.

**Strengths And Weaknesses:**

Strengths:
+ The authors simulate 16 realistic corruption types (e.g., Gaussian noise, motion blur, spatter, JPEG compression) at 5 severity levels, systematically covering key real-world scenarios that challenge FR systems. This makes the proposed benchmark relatively comprehensive.
+ The two new verification metrics proposed by the authors address some of the limitations of the existing traditional metrics.
+ The benchmark has a massive scale and considers a diverse model setting in the evaluation.
+ Experiments on real blurry images from CelebA show that synthetic corruptions in DecordFace correlate with natural corruptions, validating the framework's relevance to deployment settings. This shows the real-world transferability of the proposed framework.

Weaknesses:
- While the benchmark reveals substantial performance degradation, the paper does not discuss or consider any potential defense or robustness-enhancing methods (e.g., data augmentation, adversarial training, denoising preprocessing).
- While 16 corruptions are included, the authors explicitly exclude rain, snow, and frost due to "unrealistic appearance." However, these can occur in surveillance contexts and could have been optionally included, quantitatively justified, or at least discussed in more details.
- Though result such as mCEI is computed and reported, but no qualitative analysis (e.g., t-SNE or PCA) of how embeddings shift under corruption is provided. Visualizations could help interpret which corruptions deform feature space most severely.
- The decision to identify a proper threshold and fix it rather than use dataset-specific cross-validation (e.g., to simulate unseen shifts) is principled, but could unfairly penalize some models, especially on smaller datasets. Some discussion on how this choice affects interpretability of mVCE would strengthen the paper.